# Deep learning for enhanced risk management: a novel approach to analyzing financial reports

Xiangting Shi[1], Yakang Zhang[1], Manning Yu[2] and Lihao Zhang[3]

[1] Industrial Engineering and Operations Research Department, Columbia University, New York, United States

[2] Department of Statistics, Amsterdam Avenue New York, Columbia University, New York, United States

[3] Department of Information Engineering, Chinese University of Hong Kong, Ho Sin Hang Engineering Building, Hong Kong



## ABSTRACT

Risk management is a critical component of today's financial environment because of the enormity and complexity of data contained in financial statements. Business situations, plans, and schedule risk assessment with the help of conventional ways which involve analytical, technical, and heuristic models are inadequate to address the complex structures of the latest data. This research brings out the Hybrid Financial Risk Predictor (HFRP) model, using the convolutional neural networks (CNN) and long-short term memory (LSTM) networks to improve financial risk prediction. A combination of quantitative and qualitative ratings derived from the analysis of financial texts results in high accuracy and stability compared with the HFRP model. Evaluating key findings, the quantity of training & testing loss decreased considerably and they have their final value as 0.0013 and 0.003, respectively. According to the hypothesis, the selected HFRP model demonstrates the values of the revenue, net income, and earnings per share (EPS), and are closely similar to the actual values. The model achieves substantial risk mitigation: credit risk lowered from 0.75 to 0.20, liquidity risk from 0.70 to 0.25, market risk from 0.65 to 0.30, while operational risk is at 0.80 to 0.35. By analyzing the results of the HFRP model, it can be stated that the proposal promotes improved financial stability and presents a reliable model for the contemporary financial markets, which in turn helps in making sound decisions and improve the assessment of risks.

## INTRODUCTION

The rapid growth of financial data has brought new challenges to risk management, necessitating more sophisticated analytical approaches. Traditional methods, such as statistical models and basic machine learning algorithms, often fall short in capturing the complexity of modern financial datasets, which are characterized by their high dimensionality, multimodal nature, and dynamic changes (*Aziz & Dowling, 2019*). With the advent of deep learning (DL), there is an opportunity to significantly enhance financial

Corresponding author
Lihao Zhang, lhzhangcuhk@ieee.org

risk assessment through more robust and adaptable models (*Wu & Zhou, 2023*). DL techniques, particularly convolutional neural networks (CNN) and long-short term memory (LSTM) networks, have shown promise in handling both structured numerical data and unstructured textual data (*Leo, Sharma & Maddulety, 2019*). CNNs excel in feature extraction from text, identifying patterns that are indicative of financial risks, while LSTM networks are effective in modeling temporal dependencies in time-series data (*Craja, Kim & Lessmann, 2020*). However, existing models often focus on either textual or numerical data in isolation, limiting their ability to provide comprehensive risk assessments. This gap highlights the need for a hybrid model that can leverage both data types to offer a holistic view of financial risk. The primary aim of this study is to develop a Hybrid Financial Risk Predictor (HFRP) model that integrates the capabilities of CNN and LSTM to process both textual disclosures and numerical financial metrics simultaneously. By combining these DL architectures, the proposed model seeks to enhance the accuracy of financial risk predictions and provide deeper insights into the relationships between financial variables (*Cui & Yao, 2024*). The research question guiding this study is as follows: Can a hybrid DL model integrating CNN and LSTM networks improve the accuracy and reliability of financial risk predictions by utilizing both textual and numerical data from financial statements? To address this question, the HFRP model is designed to extract features from textual financial disclosures using CNN and capture temporal patterns in numerical data using LSTM. This joint representation aims to identify risk factors more effectively, offering a nuanced approach that traditional single-modality models cannot achieve. The study also incorporates reinforcement learning and generative adversarial networks (GANs) to enhance the model's adaptability and mitigate issues of data scarcity, further strengthening its predictive capabilities (*Fischer & Krauss, 2018*; *Gunasilan & Sharma, 2022*).

In this context, the technologies such as artificial intelligence (AI), and more particularly deep learning have appeared as a revolution (*Kim et al., 2020*; *Kraus & Feuerriegel, 2017*; *Leo, Sharma & Maddulety, 2019*; *Bertucci et al., 2022*; *Liu & Pun, 2022*; *Mai et al., 2019*). Machine learning is a subcategory of artificial intelligence where a model can find structure in patterns without being explicitly programmed to do so, deep learning is a type of machine learning that trains deep neural networks on large datasets to recognize these complex patterns. It can mainly be applied in image and speech recognition; however, its abilities in analysing financial data are still to be explored (*Mashrur et al., 2020*). Recently, the community has paid significant attention to utilizing deep learning techniques in solving various financial-related issues, such as fraud detection, trading systems, and risk estimation. The capability of deep learning models to work with and analyse unstructured data, including text from financial reports, is another good area for the improvement of risk management (*Mashrur et al., 2020*). In the context of risk management, such techniques create a possibility for delivering more adequate and timely outcomes through deep learning, which extracts the main financial data, as well as its analysis. This work further extends these improvements and proposes to create a new deep-learning method for the analysis of the financial reports and its effect on managing risks (*Matin et al., 2019*). While deploying natural language processing (NLP) and neural networks (NNs), our work

aims to overcome some of the drawbacks of classical techniques and to offer a more reliable and effective solution for today's highly growing financial industry (*Mwangi, 2024*; *Ozbayoglu, Gudelek & Sezer, 2020*).

- In traditional methods, most of the work involves the extraction of financial data, as well as its subsequent analysis, so is very tedious and often contains a degree of inconsistency.
- Traditionally, methods of risk management that are used by companies are based on the static approach and often include the historical analysis of the market conditions.
- These methods fail to handle the large volume and the increasing complexity of the financial reports hence exerting undue pressure in handling large data sets.
- This results in delaying the identification of emerging risks and their appropriate response due to the manual and heuristic nature of traditional methods.
- Heuristic models fail to reveal other characteristics of financial data so the assessment of risk is likely to be less precise.
- Such environments, full-text documents, and documents containing textual content relevant to the analysis of, for instance, financial reports are not efficiently worked through using traditional methods.
- One major drawback of the manual procedures used in the data analysis is that it can be very time-consuming and demands a lot of effort from the financial analysts.

The uncertainties in this case involve the valuing of risks and their management in the financial business since financial reports contain enormous and sophisticated information (*Peng & Yan, 2021*). Most of the risk management techniques used currently employ statistical-based models and involve analysts and experts who are unable to identify complex patterns and interrelations of this data which results in poor risk assessments and poor decision-making (*Sun & Li, 2022*). This research targets the following research problem: The limited application of deep learning in contemporary risk management (*Tan & Kok, 2024*). This current study will advance the analysis of risk by using deep learning models on text data as well as the numerical data of financial report analysis which will greatly improve the accuracy and reliability of the risk analysis (*Tian et al., 2024*). This article concerns itself with the construction of more accurate models of detection and measurement of financial risk with regards to enhancing decision and risk management in the financial industry (*Mai et al., 2019*; *Wang, Ma & Chen, 2023*).

The main purpose of this research is to propose and examine the effectiveness of the deep learning model for the improvement of risk management using financial reports. The goal of this research is to employ state-of-the-art neural networks, and natural language processing to assist in risk assessment and enhance the efficiency and reliability of the field.

- To build a deep learning model that can take structured data in the financial reports and transform and classify them.
- To combine NLP and NNs to improve the performance of risk factor detection from the financial text.

- To test the model that has been developed herein with the help of the historical financial data, and to compare its results to the results provided by the traditional risk management frameworks.
- To build a system that can improve from every new financial data that comes into the market striving for relevance and accuracy in the long run.

In that sense, this research will make the following novel contributions to the financial risk management literature: The combination of deep learning and natural language processing will give an optimal solution in the analysis of large financial data streams and flows, which is a major advantage over previous methods. This methodology shall not only enhance the effectiveness of the risk assessments but also present a better and more efficient system that is able to develop and grow with the dynamic world of finances.

1) Automated data processing: The applied research also proposes an effective solution to spot the text in the form of financial reports and, in particular, to convert unstructured into semistructured form, thus automating a large portion of the whole work on the analysis of such a type of information.
2) Enhanced risk identification: Compared to the strategies generally used in prior studies, the current work uses deep learning and NLP to offer more precise and extensive definitions of potential risk factors.
3) Scalable solution: The specifics of the flow of the proposed system are designed to enable the system to grow with this growing volume of financial data without a significant decline in efficiency or calculated results.
4) Continuous learning: The approach used in the design of the framework is that it is implemented in a flexible way which is reflected by the fact that it is capable of learning in a context in which market data change over time and is capable of learning from this new data over time.
5) Improved decision-making: The research improves decision-making in financial institutions through sufficiently timely and accurate risk evaluations which would help the institutions address new risks more effectively.

## Motivations

The key motivations of the proposed work are:

**Complexity and volume of financial data:** Huge volumes of data are presented precisely in MS Financial Statements of the contemporary period which are voluminous, extensive, detailed and compounded with unorganized information that are not easy to process and analyze by conventional techniques of analysis.

**Limitations of traditional methods:** Traditional approaches to managing risk involve the use of analysis and models which work on heuristics and take time in their processing. Neither of these methods is adequate for handling the massive amount of data that is being produced today and for the higher level of sophistication that is found in today's financial data.

**Advancements in deep learning:** Nevertheless, given that deep learning models have the potential to process and analyze big data with good accuracy then the prediction of financial risks can be enhanced. Using a combination of quantitative and qualitative data extracted from financial texts, deep learning models' risk assessment is more accurate.

**Improved risk mitigation:** Although the new approach, the Hybrid Financial Risk Predictor (HFRP) model based on CNN and LSTM networks has provided a better solution for accurate risk prediction. With the help of this model, all kinds of financial risks including credit risk, liquidity risk, market risk, and operational risk can be minimized to a great extent.

**Efficiency and accuracy:** The objective of the research is to improve the effectiveness of the risk management process by simplifying and automating it, as well as to minimize the time needed to analyze the companies' financial reports to obtain reliable results about the existence and characteristics of the potential risks. This would create a knowledge that enables the financial institutions to make right decisions and also mitigate risks effectively.

The rest of the article is organized as follows to cover all the aspects of deep learning in improving risk management with the help of financial report analysis. The Introduction draws the background that raises the extent of utilization of effective risk management mechanisms in financial markets and problems associated with the use of traditional approaches, the statement of the problem, problem formulation, and research questions and objectives. The Literature Review of this study plots the current literature on risk management, deep learning, and natural language processing to uncover research voids that this work seeks to fill. Our research's Methodology presents an overview of the steps taken to build, train, and incorporate the deep learning framework in use as well as the integration of NLP methods. In the Results and Discussions section, we report and compare the results obtained in the experiments with the traditional methods and analyze the significance of the results. Last, the Conclusions contain an overview of the main findings of the research, its implications, and possible future studies in the sphere of financial risk management.

## LITERATURE REVIEW

The application of DL in financial risk management has garnered significant interest due to its potential to enhance decision-making processes and risk mitigation strategies. This review discusses recent studies and emerging trends in utilizing DL across various financial domains, highlighting key methodologies, findings, and limitations. Historically, financial risk management has relied heavily on traditional statistical models and manual analytical approaches. However, the complexity and volume of financial data have necessitated the adoption of advanced techniques such as ML and DL. *Aziz & Dowling (2019)*, and *Leo, Sharma & Maddulety (2019)* emphasize the limitations of conventional risk management methods and advocate for integrating AI in handling complex financial data. Their studies discuss the future prospects of AI-driven risk models, highlighting the challenges related to data protection and regulatory compliance.

Detecting financial statement fraud has been a critical area of research. *Craja, Kim & Lessmann (2020)* applied CNN and recurrent neural networks (RNNs) to create robust fraud detection models. Their approach focused on recognizing patterns and anomalies within financial data, significantly improving fraud detection capabilities. This study paved the way for automated fraud identification using DL, reducing financial losses and increasing detection efficiency. The integration of DL with reinforcement learning (RL) has shown promising results in enhancing financial risk predictions. *Cui & Yao (2024)* combined DL with reinforcement learning techniques to address risk prediction challenges in supply chain management. Their method demonstrated improved risk evaluation and helped prevent disruptions in financial operations, highlighting the advantages of a dynamic decision-making framework.

LSTM networks have proven effective in analyzing time series data in financial markets. *Fischer & Krauss (2018)* utilized LSTM networks to capture temporal dependencies and nonlinear patterns in market data. Their research demonstrated enhanced market forecasting capabilities, providing more accurate investment risk assessments. The use of LSTM networks enabled dynamic risk evaluation, adapting well to volatile market conditions. Machine learning techniques have been increasingly adopted in the banking sector for risk management. *Gunasilan & Sharma (2022)* explored the use of neural networks and decision trees as alternatives to traditional risk assessment models. Their findings highlighted the effectiveness of these techniques in improving risk prediction accuracy, although challenges in real-world implementation and regulatory compliance remain.

*Kraus & Feuerriegel (2017)* investigated the use of transfer learning for decision support in financial risk management. By applying deep neural networks to financial disclosures, their approach enhanced the quality of decision-making in corporate finance. Transfer learning facilitated the adaptation of pre-trained models to specific financial datasets, improving risk analysis accuracy. Several studies have focused on using DL for predicting financial distress. *Matin et al. (2019)* employed CNN and RNNs to analyze textual segments from annual financial reports, identifying early indicators of potential financial failure. The use of NLP allowed for the extraction of nuanced risk indicators from unstructured textual data, contributing to proactive financial risk management.

Comprehensive surveys by *Mashrur et al. (2020)* have reviewed the application of machine learning in financial risk control. Their work identified key trends in the adoption of complex neural network architectures like CNN and RNNs. These studies emphasized the transition from traditional models to deeper learning approaches, underscoring the enhanced predictive accuracy achieved through hierarchical and temporal data analysis. *Liu & Pun (2022)* developed a two-step supervised learning approach to measure systemic risk and predict corporate bankruptcy. Their model integrated textual and quantitative data, capturing the interconnectedness of financial entities. This approach allowed for more precise and timely evaluations of systemic risks, addressing the complexities of financial market dynamics. Attention mechanisms have been utilized to enhance the precision of fraud detection models. *Wang (2024)* proposed an attentive method for

identifying fraudulent activities in multimodal financial data. Their model effectively integrated textual and numerical features, improving the detection accuracy and robustness of financial risk predictions.

Incorporating multi-scale learning techniques, *Yue (2024)* introduced a hierarchical DL approach for financial risk management in higher education institutions. This method improved risk prediction accuracy and supported strategic financial planning, showcasing the versatility of DL architectures in handling diverse financial contexts. *Zhao et al. (2024)* explored DL applications for enhancing enterprise risk management systems during digital transformation. Their model combined deep neural networks with reinforcement learning, providing scalable and flexible solutions to meet the demands of evolving digital markets. This approach emphasized the adaptability of advanced neural networks in enterprise-level decision-making. *Zubair, Bhatti & Meghji (2024)* applied DL models to analyze the financial health of the banking sector in Pakistan. Their research demonstrated the efficiency of deep neural networks and RNNs in predicting economic trends, aiding in policy-making and financial stability.

The literature highlights significant advancements in using DL for financial risk management, with applications ranging from fraud detection and market forecasting to systemic risk prediction. However, there remain challenges in model interpretability, data quality, and scalability, particularly when integrating explainable AI techniques. More research is needed to develop robust, interpretable models capable of real-time risk supervision and decision support across diverse financial environments. Table 1 shows the comparison of techniques, research focus, findings, and limitations in financial risk management.

Although prior research provides a rich body of study on the usage of DL in FRM more specific attention is lacking with regard to the explanation techniques utilized to propel the accountability and interpretability of the models. However, more work is required to understand and prove deep learning models' efficiency and applicability to various financial contexts and situations primarily concerning real-time risk supervision and effective decision-making systems.

From the reviewed literature we identified a wide variety of research studies. Applications of AI are: using textual analytics to predict companies' financial distress, using neural networks to improve the efficiency of fraud detection, and using LSTM networks to boost the precision of market forecasts. Furthermore, there is research carried out to understand the capabilities of deep learning in the realm of compliance and the extent of its influence in redesigning approaches to risk management for financial organisations. These contributions support and map the current state of the art and application of deep learning in financial risk management indicating that, it is a promising field that is gaining grounds in financial risk, and there's improved room for future studies and practice.

## METHODOLOGY

This section describes the general methodology involved in the research together with specifics on the dataset, text preprocessing, and the developed novel deep learning model

**Table 1  Comparison of techniques, research focus, findings, and limitations in financial risk management.**

| Author(s) | Techniques | Research focus | Key findings and applications | Limitations |
|---|---|---|---|---|
| *Tan & Kok (2024)* | Explainable AI | Risk classification in financial reports | Enhanced transparency in risk classification models for financial reports, improving the interpretability of decisions. | Limited scalability of explainable AI techniques to complex financial datasets. |
| *Mashrur et al. (2020)* | Machine learning | Survey of applications in financial risk management | Identified advancements and challenges in machine learning applications for financial risk management. | Lack of detailed comparative analysis between different machine learning techniques. |
| *Zhao et al. (2024)* | Deep learning | Financial investment risk prediction | Improved accuracy in financial investment risk prediction using deep learning models. | Dependency on data quality and availability for training deep learning models. |
| *Gunasilan & Sharma (2022)* | Machine learning | Alternative analysis tools in banking sectors | Explored alternative machine learning tools for risk management in banking sectors. | Limited discussion on real-world implementation challenges and regulatory implications. |
| *Matin et al. (2019)* | Deep learning | Predicting financial distress | Predicted financial distress through textual data analysis in annual reports. | Reliance on textual data quality and relevance in predicting financial distress accurately. |
| *Fischer & Krauss (2018)* | LSTM networks | Financial market predictions | Enhanced financial market predictions using LSTM networks to capture temporal dependencies. | Sensitivity to hyperparameter tuning and model architecture choices in LSTM networks. |
| *Ozbayoglu, Gudelek & Sezer (2020)* | Deep learning methods | Synthesis of applications across financial domains | Synthesized deep learning applications across financial domains, highlighting methodological advancements. | Generalization of findings across diverse financial applications without specific use cases. |
| *Aziz & Dowling (2019)* | AI and machine learning | Risk modeling and operational efficiencies | Explored AI-driven improvements in risk modeling and operational efficiencies in financial institutions. | Challenges related to data privacy and ethical considerations in deploying AI in financial contexts. |
| *Wang (2024)* | Attention mechanisms | Fraud detection in multimodal financial data | Developed attentive fraud detection methods for multimodal financial data, improving detection accuracy. | Complexity in integrating attention mechanisms with existing fraud detection frameworks. |
| *Mwangi (2024)* | Machine learning | Optimization of risk management strategies | Investigated machine learning's impact on risk management strategies, optimizing predictive modeling. | Case-specific findings may not generalize to broader financial institutions without further validation. |

for financial risk prediction. There are the following steps in the ontological development of the proposed HFRP model as shown in Fig. 1. The respective automating process consists of data accumulation of financial data and data pre-processing steps including, tokenization, stop word removal, and stemming or lemmatization. The obtained preprocessed data is then passed through the vectorization process and then trained using the CNN-LSTM model. It is followed by the training process; however, at first, there are three dense layers being implemented. Throughout the training process, calculation and optimization of the loss occur step by step. The trained model is then tested in terms of various metrics and the obtained results are analyzed to analyze the efficiency of the risk management strategies.

The core objective of this research is to develop a HFRP that utilizes DL neural networks for enhanced financial risk analysis. The proposed HFRP model combines CNN and

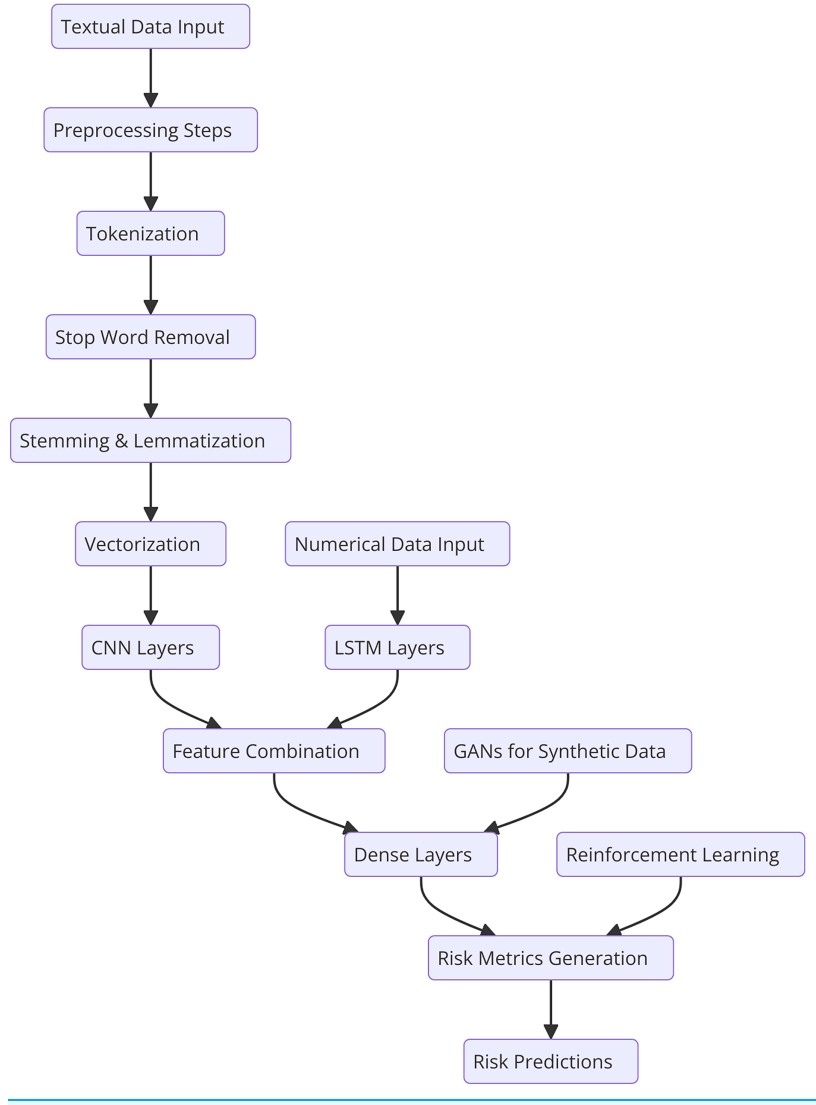

**Figure 1 Detailed workflow for HFRP.**

LSTM networks to handle both textual and numerical data from financial reports. This hybrid architecture aims to leverage the strengths of each neural network type:

- **CNNs** excel at extracting meaningful features from unstructured text data, such as financial disclosures.
- **LSTMs** are effective at capturing temporal dependencies within sequential numerical data, like time series of financial metrics.

The integration of these DL techniques addresses the complexity and high-dimensional nature of financial datasets, providing a comprehensive approach to risk prediction. This model aligns with the scope of the journal, which focuses on innovative applications of AI in financial risk management. While prior studies have explored DL applications in

financial risk prediction, a significant knowledge gap remains in integrating multimodal data sources (*i.e.*, combining both text and numerical data) within a single predictive framework. Traditional risk assessment models typically rely on either qualitative analysis (text-based) or quantitative metrics (numerical-based), limiting their predictive power and failing to capture the full context of financial reports.

*How can a hybrid DL model combining CNN and LSTM improve the accuracy and reliability of financial risk predictions by leveraging both textual and numerical data from financial reports?*

This research question is crafted to address the identified gap in the literature. By proposing a hybrid approach, the study aims to overcome the limitations of existing models that handle only a single type of data. The research hypothesizes that combining textual analysis with numerical time series analysis will yield better predictive performance, providing a holistic view of a company's financial health and associated risks.

## Knowledge gap and contribution

Existing financial risk models predominantly employ statistical methods or single-modality neural networks, such as using only CNN for text analysis or LSTM for numerical data. These methods have inherent limitations:

1) **Inadequate text analysis:** Traditional models often fail to extract semantic information from complex, unstructured text data present in financial disclosures.
2) **Lack of temporal awareness:** Models using only numerical data typically miss out on the contextual nuances captured in textual information, leading to suboptimal risk assessments.
3) **Scalability issues:** Many models are designed for specific financial contexts and may not generalize well across different industries or economic conditions.

The HFRP model addresses these issues by:

- Combining the strengths of CNN and LSTM to process both textual and numerical data simultaneously.
- Utilizing advanced NLP techniques for feature extraction from financial disclosures, enhancing the model's ability to interpret qualitative risk factors.
- Implementing a scalable framework that adapts to diverse financial datasets, improving the generalizability of the predictions.

This integrated approach fills a critical gap in the current state of financial risk analysis by providing a unified model capable of capturing intricate relationships within multimodal data. It aims to set a new standard for predictive modeling in financial risk management, with broader implications for decision-making processes in financial institutions.

## Computing infrastructure

The system configuration includes Ubuntu 20.04 LTS as the operating system, providing a stable and efficient platform for various tasks. The hardware is equipped with an Intel Core i7-9700K CPU running at 3.60 GHz, complemented by 32 GB of RAM, which ensures smooth multitasking and high performance. For graphical processing, it utilizes an NVIDIA GeForce RTX 2070. On the software side, the setup includes Python 3.8 for programming, Jupyter Notebook for interactive computing, and Anaconda Distribution for managing packages and environments.

## Model description

The proposed model is based CNN and LSTM networks.

*Convolutional neural networks (CNN)*: CNN are primarily used for feature extraction from text data. They are effective in identifying and learning patterns within the data through convolutional layers. The CNN part of the model consists of multiple convolutional layers followed by pooling layers, which help in reducing the dimensionality of the data while preserving important features.

*Long short-term memory (LSTM) networks*: LSTM are utilized for capturing temporal dependencies in the data. They are particularly useful for sequential data and can effectively handle long-term dependencies. The LSTM network includes multiple LSTM cells, each capable of maintaining information over long periods, which is essential for understanding the context in financial texts.

*Integration of CNN and LSTM:* The integration of CNN and LSTM networks leverages the strengths of both models. CNN are highly effective for feature extraction from textual data, while LSTM excel at capturing temporal dependencies. The combined model, referred to as HFRP, is specifically designed to process both quantitative and qualitative aspects of financial data, providing a more comprehensive risk assessment.

*Evaluation:* The performance of the model is evaluated using standard metrics such as accuracy, precision, recall, and F1-score to ensure its reliability. Additionally, the model is validated on a separate dataset to fine-tune parameters and prevent overfitting.

Figure 1 shows the workflow begins with two distinct input streams: textual data and numerical data. The textual data undergoes preprocessing, which includes tokenization, stop word removal, stemming, and lemmatization to prepare it for analysis. This preprocessed text is then vectorized and passed through CNN layers to extract meaningful semantic features. Simultaneously, the numerical data, typically structured financial metrics, is fed into LSTM networks, which capture temporal dependencies and trends inherent in the data. The features extracted from the CNN and LSTM layers are then combined into a unified feature representation, enabling the model to leverage insights from both textual and numerical data. This joint representation is processed through a series of dense layers for further refinement and decision-making. To enhance model robustness and mitigate data scarcity, GANs are incorporated, generating synthetic data to improve training quality. Additionally, RL is applied to adapt the model dynamically based on evolving market conditions, ensuring continuous improvement in predictions. Finally,

the refined features are used to generate risk metrics, culminating in accurate and comprehensive financial risk predictions. This integrated approach allows for a holistic analysis of financial risks, leveraging the strengths of multiple deep learning techniques.

## Comprehensive risk prediction in financial reports using hybrid deep learning models

Introduce a deep learning system that integrates attributes of CNN and LSTM to dental with textual and numerical data from financial statements. The objective is to formally define a model that is capable of identifying financial risks and how different variables may depend on one another within the data. The proposed Hybrid Financial Risk Predictor (HFRP) model integrates CNN and LSTM networks to process both textual and numerical data from financial statements. The input data representation involves two components: $X$, which is a matrix capturing numerical features of size $n \times m$, and $T$, a tensor of dimensions $n \times p \times q$ for the textual data extracted from financial reports. The CNN is utilized for feature extraction from the textual data. It applies convolution operations using filter weights ($W$conv) and biases ($b$conv), followed by an activation function ($\sigma$), resulting in feature maps ($F_i$ CNN). This process captures essential patterns and semantic information from the text data, which are critical for identifying potential risk indicators.

**1. Input data representation:**

$$\mathbf{X} \in \mathbb{R}^{n \times m}, \quad \mathbf{T} \in \mathbb{R}^{n \times p \times q} \tag{1}$$

**2. Feature extraction:**

$$\mathbf{F}_i^{\text{CNN}} = \sigma(\mathbf{W}_{\text{conv}} * \mathbf{T}_i + \mathbf{b}_{\text{conv}}) \tag{2}$$
$$\mathbf{h}_t = \phi(\mathbf{W}_h \cdot \mathbf{x}_t + \mathbf{U}_h \cdot \mathbf{h}_{t-1} + \mathbf{b}_h) \tag{3}$$

**3. Combined feature representation:**

$$\mathbf{F}_i = \left[\mathbf{F}_i^{\text{CNN}}, \mathbf{h}_T\right] \tag{4}$$

**4. Objective function (Eq. (5)):** This represents the objective function used in the HFRP model. Its components ensure that the model balances minimizing prediction error while avoiding overfitting and improving model sparsity:

$$\min_{\theta} \mathcal{L} = \frac{1}{n} \sum_{i=1}^{n} \left[ \alpha(y_i - \hat{y}_i)^2 + \beta\left(\sum_{j=1}^{k} \theta_j^2\right) + \gamma \sum_{j=1}^{m} |\theta_j| \right] \tag{5}$$

**5. Prediction function:**

$$\hat{y}_i = f(\mathbf{F}_i) = \mathbf{W}_{\text{out}} \cdot \mathbf{F}_i + \mathbf{b}_{\text{out}} \tag{6}$$

This formulation jointly uses CNN for text and LSTM for numerical data thus formulating a joint feature representation. The objective function also consists of mean

squared error, L2 regularization, and L1 regularization to increase the optimization's complexity.

## Dynamic financial risk forecasting using reinforcement learning and deep learning integration

Propose an advanced risk prediction model that may combine deep reinforcement learning (DRL) with traditional deep learning to improve the forecast with new data. The model will adjust techniques of risk management about a dynamic market environment.

**1. State representation:**

$$s_t = [\mathbf{X}_t, \mathbf{T}_t] \tag{7}$$

**2. Action space:**

$$a_t \in \{\text{buy, sell, hold}\} \tag{8}$$

**3. Reward function:**

$$r_t = \omega_1 \cdot \text{Accuracy}(\hat{y}_t, y_t) + \omega_2 \cdot \text{Return}(\text{Portfolio}_t) \tag{9}$$

**4. Policy network:**

$$\pi(a_t|s_t; \theta) = \text{softmax}(\mathbf{W}_\pi \cdot s_t + \mathbf{b}_\pi) \tag{10}$$

**5. Value function:**

$$V(s_t; \phi) = \mathbf{W}_V \cdot s_t + \mathbf{b}_V \tag{11}$$

**6. Objective function:**

$$\min_\theta \mathcal{L}_\pi = -\mathbb{E}[\log \pi(a_t|s_t; \theta) \cdot (r_t + \gamma V(s_{t+1}) - V(s_t))] \tag{12}$$

$$\min_\phi \mathcal{L}_V = \frac{1}{2}\mathbb{E}[(r_t + \gamma V(s_{t+1}; \phi) - V(s_t; \phi))^2] \tag{13}$$

This problem formulation is a coupling of DRL with deep learning that makes it a dynamic model that can learn from new financial data a current feed. That is, the policy and value networks are trained to acquire the best risk management solutions at a level of accuracy desired while being a good investment.

For numerical data, the LSTM network captures temporal dependencies and trends using its hidden states. The hidden state at time t, denoted as $h_t$, is computed using the input data ($x_t$), previous hidden state (***ht − 1***), and respective weight matrices ($W_h$ and $U_h$). This mechanism allows the model to retain relevant information over long sequences, making it effective for analyzing time series data like financial metrics. The extracted features from both CNN and LSTM are then combined into a unified feature vector ($F\_i$), representing a comprehensive joint feature set that integrates insights from both text and numerical data.

The objective function for the model is designed to minimize prediction error while also incorporating regularization terms to prevent overfitting. The loss function includes three components: the mean squared error between the actual $(y_i)$ and predicted values $(y^i)$, an L2 regularization term $(\beta \sum_\theta^{j2} j)$ to penalize large weight values, and an L1 regularization term $(\gamma \Sigma \theta j)$ to encourage sparsity in the model parameters. Finally, the prediction function applies a linear transformation using output weights $(W_{out})$ and biases $(b_{out})$, generating the final risk prediction $(y^i)$ based on the combined feature representation.

## Multi-modal financial risk analysis using GANs

Propose multi-modal risk assessment financial risk GAN for generating synthetic data for increasing the accuracy of the forecasts. The model is expected to solve problems of data deficit and at the same time improve the efficiency of risk modelling.

**1. Generator network:**

$$\mathbf{G}(z; \theta_G) = \sigma(\mathbf{W}_G \cdot z + \mathbf{b}_G) \tag{14}$$

**2. Discriminator network:**

$$D(\mathbf{x}; \theta_D) = \sigma(\mathbf{W}_D \cdot \mathbf{x} + \mathbf{b}_D) \tag{15}$$

**3. Adversarial loss:**

$$\min_{\theta_G} \max_{\theta_D} \mathcal{L}_{\text{GAN}} = \mathbb{E}_{\mathbf{x} \sim p_{\text{data}}}[\log D(\mathbf{x}; \theta_D)] + \mathbb{E}_{z \sim p_z}[\log(1 - D(\mathbf{G}(z; \theta_G); \theta_D))] \tag{16}$$

**4. Risk prediction network:**

$$\hat{y} = f(\mathbf{x}; \theta_R) = \sigma(\mathbf{W}_R \cdot \mathbf{x} + \mathbf{b}_R) \tag{17}$$

**5. Objective Function for risk prediction:**

$$\min_{\theta_R} \mathcal{L}_R = \frac{1}{n} \sum_{i=1}^{n} \left[ \alpha \left( y_i - \hat{y}_i \right)^2 + \beta \left( y_i - \hat{y}_{i,\text{synthetic}} \right)^2 + \lambda \left( \sum_{j=1}^{k} \theta_j^2 \right) \right] \tag{18}$$

The model incorporates a GAN framework, which consists of a generator network and a discriminator network. The generator (refer to Eq. (14)) receives a random noise input vector and transforms it using a series of weight matrices, biases, and activation functions. This process produces synthetic data that aims to resemble real financial data. On the other side, the discriminator (refer to Eq. (15)) evaluates both real and generated data. It applies a similar transformation process with its own weights and biases, outputting a probability score that indicates whether the input is real or synthetic. The training process for the GAN involves an adversarial loss function (refer to Eq. (16)). Here, the generator attempts to minimize its loss by creating realistic data, while the discriminator tries to maximize its ability to correctly distinguish between real and synthetic data. This adversarial setup ensures that the generator continually improves the quality of the synthetic data. For predicting financial risk, the model includes a risk prediction network (refer to Eq. (17)),

which applies a linear transformation to the combined input features using a set of weights and biases, followed by an activation function. The network generates the predicted risk values based on these transformations. The model's overall objective function (refer to Eq. (18)) minimizes the prediction error, incorporating regularization terms to prevent overfitting. This loss function includes three components: the mean squared error between actual and predicted values, an error term for discrepancies in synthetic data predictions, and regularization penalties to ensure model stability. Together, these elements contribute to an effective risk prediction and analysis framework.

This formulation employs the use of GANs to synthesise financial data and reduces the problem of data deficiency as well as enhances the accuracy of risk prediction. The adversarial loss guarantees the synthesized data to be realistic while the combined loss improves the stability of the risk prediction network. Additionally, the framework includes a GAN component for synthesizing financial data, which addresses issues of data scarcity and enhances model robustness. The GAN consists of a generator network (G), which produces synthetic data, and a discriminator network ($D$ D), which differentiates between real and generated data. The adversarial loss function optimizes both networks, ensuring that the generated data closely mimics the real data. The risk prediction network leverages the synthetic data alongside real data, minimizing a combined loss function that includes both the error in predicting real data and the discrepancy between real and synthetic predictions. This multi-modal approach allows the model to generalize better and provide more accurate financial risk assessments, even when faced with limited or incomplete data. In summary, the HFRP model uses a hybrid architecture combining CNN and LSTM to extract and analyze features from both textual and numerical data. The dynamic integration of reinforcement learning adapts to market changes, while the GAN module enhances data quality and model reliability. This comprehensive system aims to provide a robust solution for financial risk prediction, leveraging the strengths of DL techniques to offer a nuanced and effective approach to risk management.

## Dataset description

This work's data set is made up of a large number of documents containing financial reports of different companies. The text and numerical financial information are also provided in the disclosures. Table 2 contains a detailed description of the dataset:

The dataset spans over ten years, encompassing various economic conditions, which allows for robust model training and validation.

In Fig. 2 (bottom-right scatter plot), the increased number of data points per company likely reflects the presence of multiple financial entries over a period of time. Instead of representing a single point for each company, the scatter plot aggregates data across different fiscal years or quarterly reports, capturing variations in total assets and total liabilities for each firm. This approach helps visualize the financial trends and stability of the companies over time, providing a more comprehensive overview of their financial positions rather than a static snapshot. The recurrent entries for the same companies

**Table 2 Feature description.**

| Feature | Description | Category |
|---|---|---|
| Company name | Name of the company | Categorical |
| Report date | Date of the financial report | Date |
| Revenue | Total revenue reported in USD | Numerical |
| Net income | Net income reported in USD | Numerical |
| Earnings per share (EPS) | Earnings per share in USD | Numerical |
| Total assets | Total assets reported in USD | Numerical |
| Total liabilities | Total liabilities reported in USD | Numerical |
| Operating income | Operating income reported in USD | Numerical |
| Cash flow from operations | Cash flow from operating activities in USD | Numerical |
| Textual disclosures | Full text of financial disclosures | Text |

enhance the depth of analysis by highlighting temporal changes in their financial metrics.

Figures 2 and 3 illustrating data distribution, such as histograms and scatter plots, are not merely for showing basic statistics; they are essential for uncovering the underlying financial trends. For example, the histograms of revenue, net income, earnings per share (EPS), and total assets reveal important insights into the central tendency, skewness, and outliers in the financial data. The presence of outliers can indicate firms with extraordinary performance or extreme financial distress, both of which are critical for assessing risk. Additionally, scatter plots showing the relationship between total assets and total liabilities help identify potential liquidity issues. A linear trend suggests stable growth, while deviations from this trend can highlight companies at risk of financial imbalance. By delving into these patterns, we gain a deeper understanding of the financial health of the dataset, allowing for better model calibration.

In Fig. 2, their respective revenue, net income, EPS, as well as the distribution of total assets against the total liability of the identified companies in the dataset. The histograms give information about the distribution and measure of the central tendency of these financial metrics and the scatter plot shows how these firms' total assets fare against their total liabilities. Figure 3 shows bar plots of operating income, cash from operating activities, total assets, and total liabilities. Such plots are useful for analyzing patterns and dispersion of the data which is essential for constructing and assessing the predictive models applied in this research.

Figure 4 shows the most important feature distribution. Instead of simply listing the financial metrics with their importance scores, this section should connect these scores to practical implications. The feature importance analysis, particularly using permutation importance, identifies key predictors for financial risk, such as net income, total assets, and total liabilities. These features rank highest due to their strong correlation with a company's ability to meet its financial obligations and withstand market fluctuations. The emphasis on net income reflects its role as a direct measure of profitability, while total

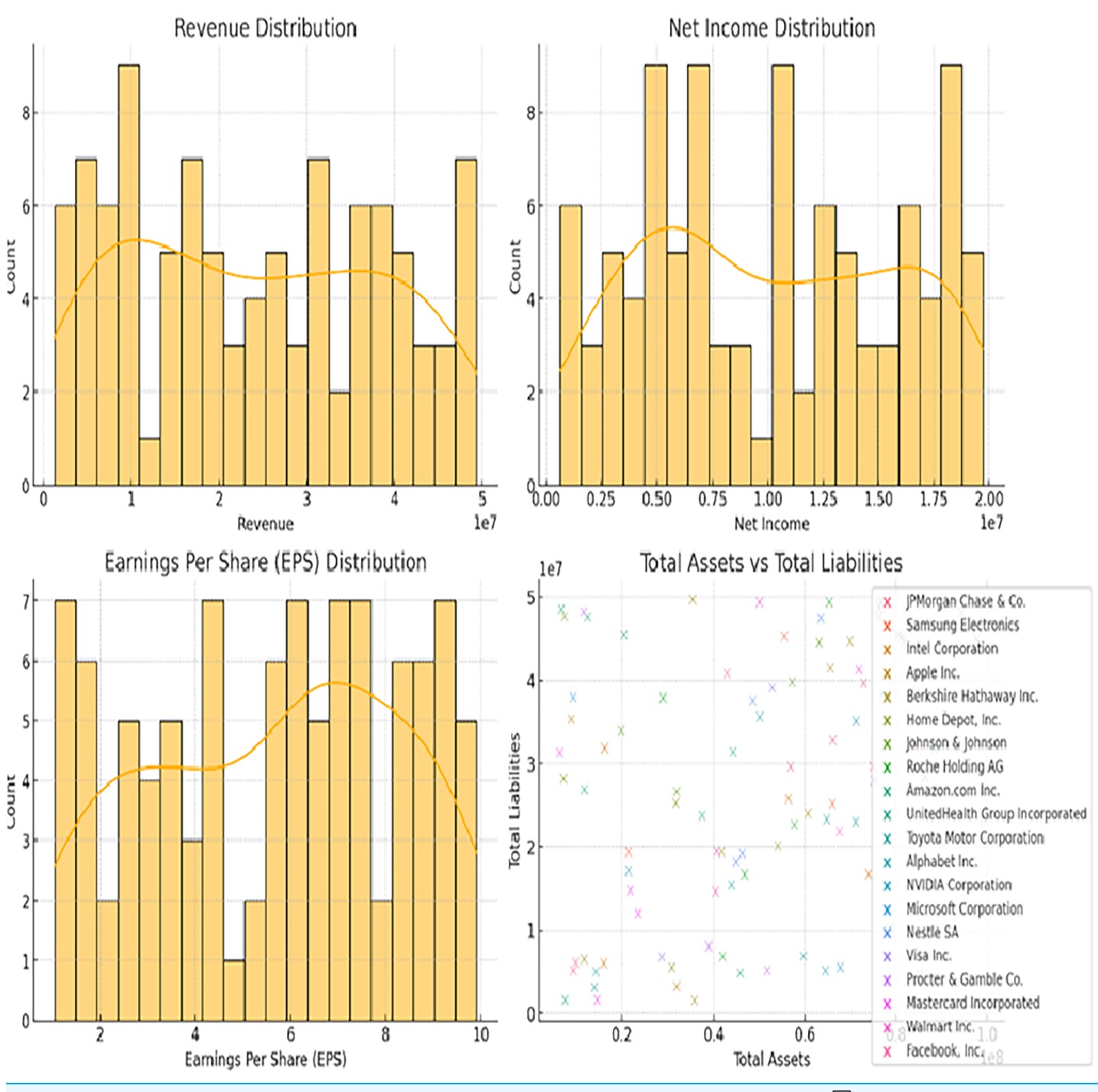

**Figure 2** Distributions of revenue, net income, EPS, and total assets *vs* total liabilities.

assets and liabilities are critical indicators of a company's financial structure and risk exposure. Understanding these key features helps refine the model's focus, improving its predictive performance and aligning it with financial realities.

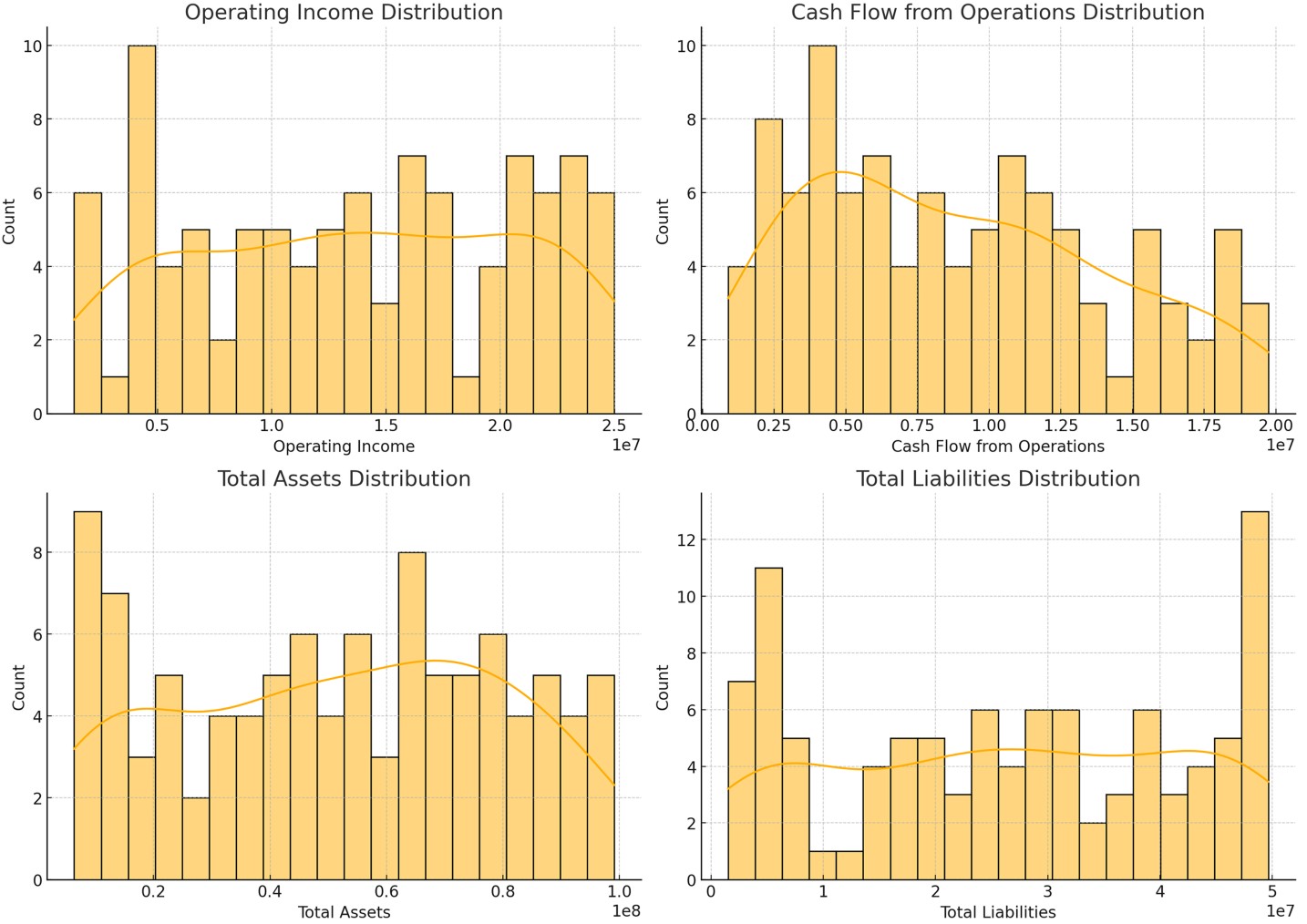

**Figure 3** **Distributions of operating income, cash flow from operations, total assets, and total liabilities.**

## Data preprocessing

The data is preprocessed to extract meaningful features. The following steps are carried out during data preprocessing:

### Text preprocessing

Text preprocessing is crucial for extracting meaningful features from the textual disclosures in financial reports. Figure 5 shows the analysis of texts. The following steps are implemented:

1) **Tokenization:** Splitting the text into individual words or tokens.

2) **Stop word removal:** Eliminating common words that do not contribute to the analysis, such as "and", "the", *etc*.

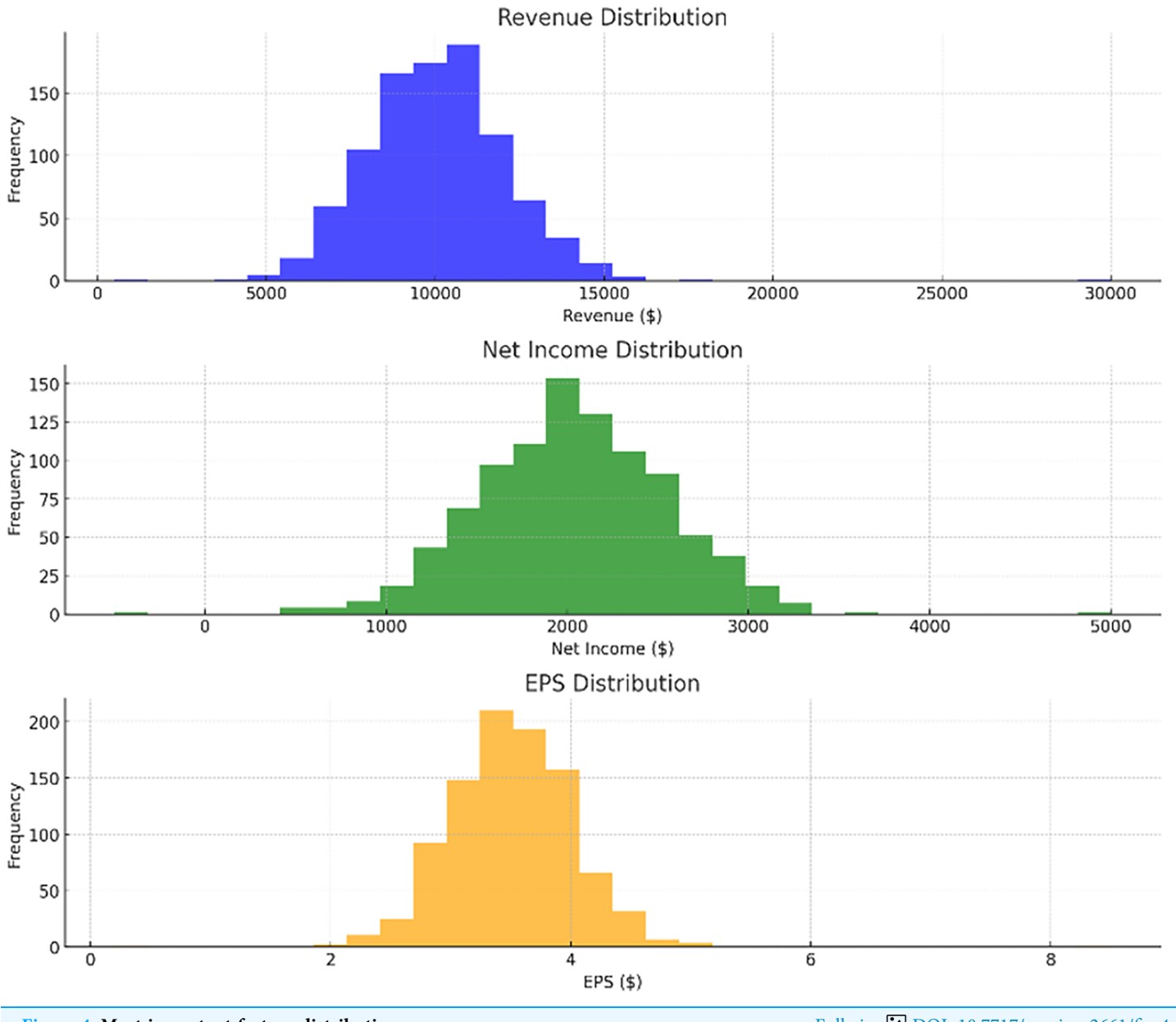

**Figure 4 Most important feature distribution.**

3) **Stemming and lemmatization:** Reducing words to their base or root form to ensure uniformity.

4) **Vectorization:** Converting the text into numerical vectors using techniques like TF-IDF or Word2Vec.

### Data splitting

In the step the dataset is divided into training, validation, and test sets to ensure the model can generalize well to new, unseen data. The analysis of texts is shows in Fig. 5.

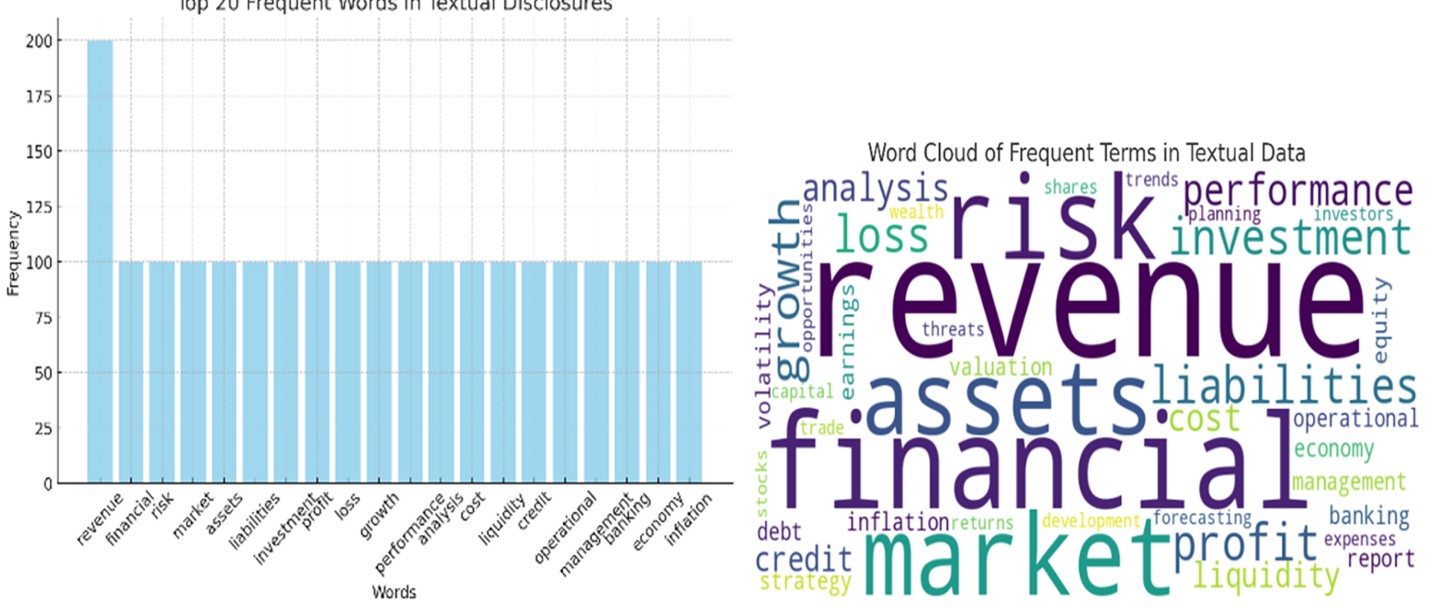

**Figure 5** Analysis of texts.

### Combining features

In this stage of data preprocessing, we've done the integration of the features extracted from both the textual and numerical data for a comprehensive analysis.

The textual disclosures' cumulative occurrence of the top 20 words is illustrated in Fig. 6. This helps in comprehending some of the frequently used terms that appear in financial reports with the aid of which it may be important for the next steps of analysis and modeling.

### Correlation analysis

It is necessary to remember that every single financial characteristic has its relationships with the other ones, and thus to determine the interconnections between various analytics is crucial for constructing effective math models. Figure 7 shows the results of the correlation test between the financial measures adopted in this study. The correlation matrices of all the financial ratios have been depicted in Fig. 7 for the convenience of understanding. It is also useful in feature selection and model building as it assists in the determination of the relations and interdependencies of the metrics.

The correlation analysis reveals critical interdependencies among financial metrics that are vital for achieving the research objectives. For instance, a strong positive correlation between revenue and net income highlights how revenue growth consistently translates into profitability, underlining its importance as a key feature in risk assessment. Similarly, a linear relationship between total assets and total liabilities indicates proportional scaling, which is beneficial for financial stability but raises liquidity concerns if liabilities grow disproportionately. Negative correlations, such as between operating income and total

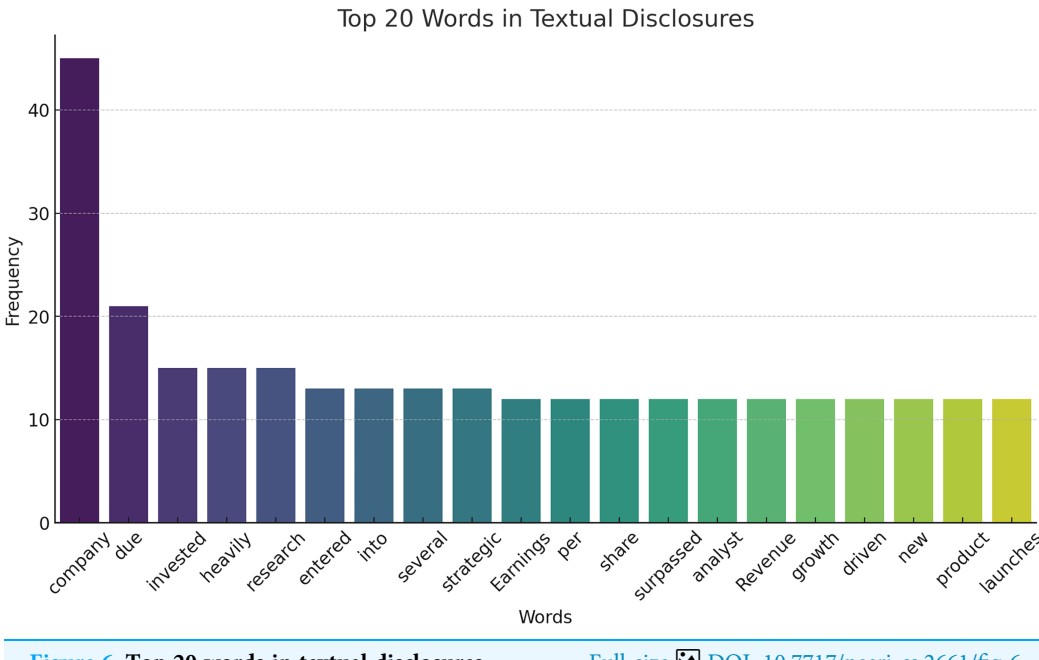

**Figure 6** **Top 20 words in textual disclosures.**

liabilities, signal potential red flags for firms with high debt struggling to maintain operational profitability. These insights directly support the design of the HFRP model by guiding feature selection and prioritization, ensuring the model focuses on metrics that significantly influence financial stability and risk. Additionally, identifying weak or anomalous correlations helps the model flag firms with unbalanced financial structures, enhancing its predictive accuracy. Correlation analysis also validates the use of LSTM networks for temporal dependencies and supports the interpretability of the model, providing stakeholders with clear, data-driven justifications for predictions. This alignment between data characteristics and model design ensures the framework's robustness and reliability in real-world risk management scenarios.

## Feature importance metrics

To establish the relevance of the various financial metrics in the computation of the risk we used permutation importance. Figure 8 exhibits the details of the feature importance of the HFRP model. The relative importance of each predictor can be seen in Fig. 8: The most important test has Net Income as the measure, while total assets and total liabilities come next. They have the highest importance in the model, which overall suggests that they are probably the most relevant features in assessing financial risk.

## Principal component analysis and clustering

Below is the cluster analysis based on principal component analysis (PCA) on the following financial metrics after processing the data collected, the following is the conclusion. It also becomes easy to find clusters of companies with characteristics that differentiate them financially. Figure 9 presents the clustering of the financial metrics in

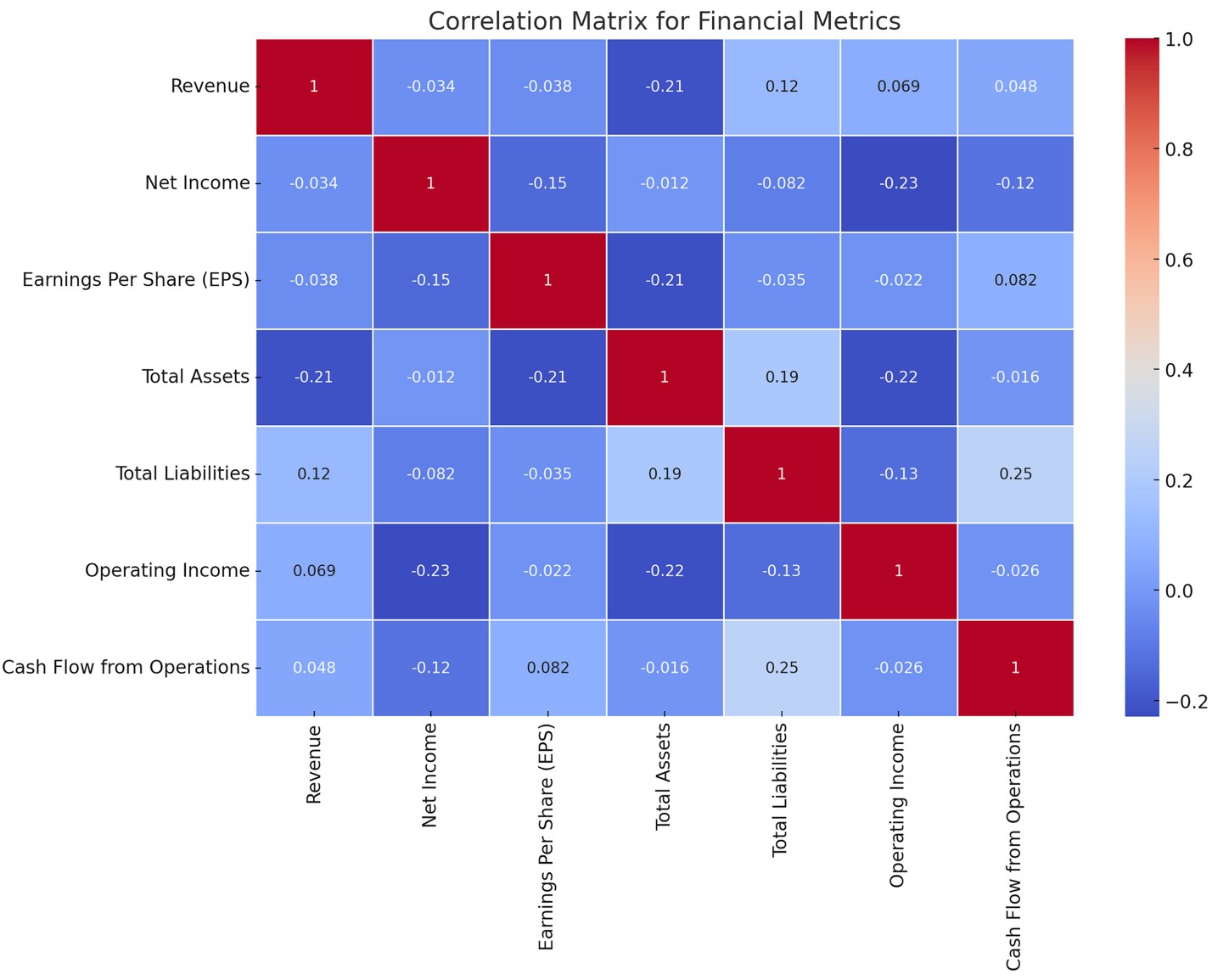

Figure 7 **Correlation matrix for financial metrics.**

which they are distinguished in terms of color at each cluster. The companies in each cluster are listed below:

Figure 9 shows the clustering of financial metrics, with each color representing a different cluster. The initial description of PCA lacked depth and did not justify its usage. PCA is employed to reduce the dimensionality of the financial data, making it more manageable for clustering without losing significant information. By transforming the original features into a set of uncorrelated principal components, PCA captures the variance and underlying structure in the data. The first few principal components usually explain a substantial portion of the variance, highlighting the key drivers of financial performance. In this study, PCA not only aids in visualization but also helps identify

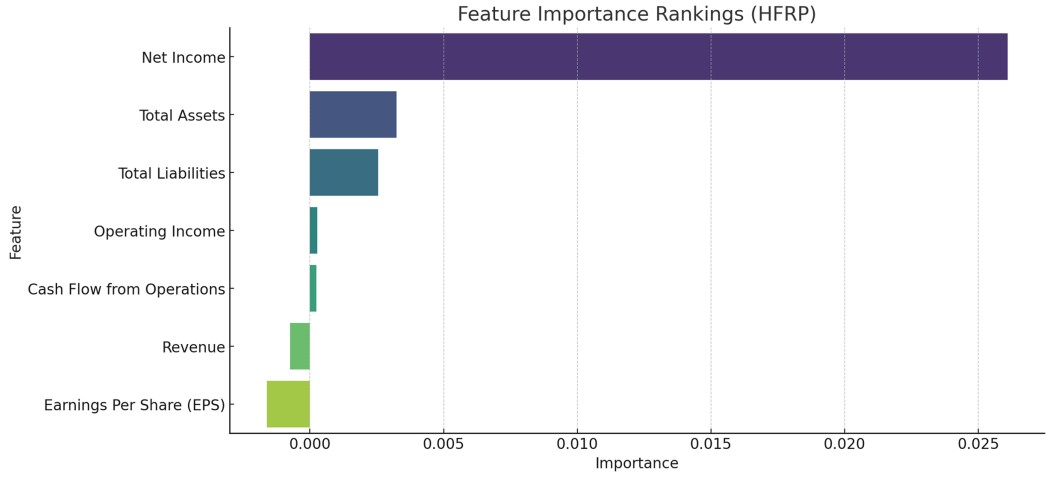

**Figure 8 Feature importance rankings for the HFRP model.**

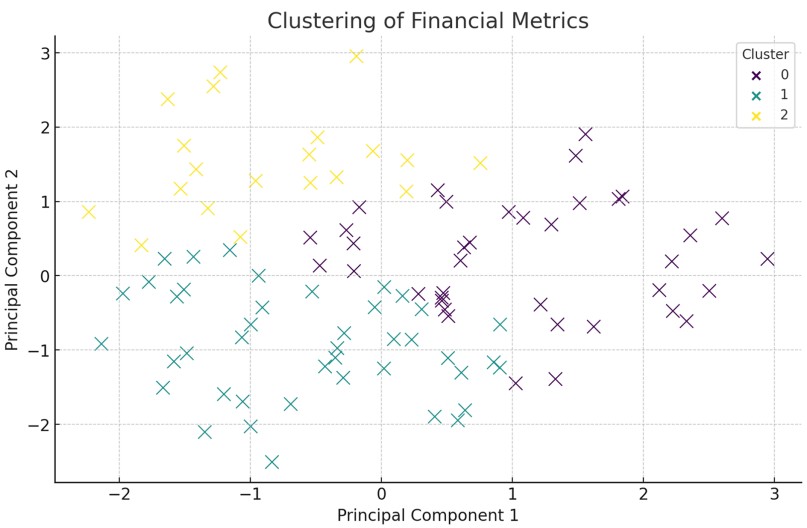

**Figure 9 Clustering of financial metrics using PCA.**

patterns related to financial stability and risk, such as distinguishing between firms with strong asset growth *vs* those relying heavily on liabilities. This dimensionality reduction is critical for effective clustering and subsequent analysis. The discussion on PCA and clustering should emphasize how the identified clusters represent distinct financial risk profiles and inform targeted mitigation strategies. Each cluster, based on key financial metrics like net income, total assets, and liabilities, reflects groups of firms with shared financial characteristics. For example, a cluster with low net income and high liabilities could indicate firms at high risk of financial distress, necessitating strategies focused on debt reduction and liquidity management. Conversely, clusters with high revenue and assets may represent stable firms with minimal risk, requiring less aggressive mitigation. By analyzing these differences, the clustering approach provides actionable insights, enabling tailored interventions for each risk group. This targeted strategy not only optimizes

**Table 3 Companies in each cluster.**

| Cluster 0 | Cluster 1 | Cluster 2 |
|---|---|---|
| JPMorgan Chase & Co. | Pfizer Inc. | Oracle Corporation |
| Samsung Electronics | Cisco Systems, Inc. | Broadcom Inc. |
| Intel Corporation | Exxon Mobil Corporation | Qualcomm Inc. |
| Apple Inc. | Chevron Corporation | Texas Instruments Incorporated |
| Berkshire Hathaway Inc. | Pepsico, Inc. | Salesforce.com, Inc. |
| Home Depot, Inc. | The Coca-Cola Company | PayPal Holdings, Inc. |
| Johnson & Johnson | Bank of America Corporation | Netflix, Inc. |
| Roche Holding AG | Wells Fargo & Company | Adobe Inc. |
| Amazon.com Inc. | AT&T Inc. | ASML Holding N.V. |
| United Health Group Incorporated | Comcast Corporation | Tesla, Inc. |
| Toyota Motor Corporation | Merck & Co., Inc. | General Motors Company |
| Alphabet Inc. | Abbott Laboratories | Ford Motor Company |
| NVIDIA Corporation | AbbVie Inc. | Honda Motor Co., Ltd. |
| Microsoft Corporation | L'Oréal S.A. | Bayerische Motoren Werke AG (BMW) |
| Nestle SA | Unilever N.V. | Daimler AG |
| Visa Inc. | Prologis, Inc. | Nissan Motor Co., Ltd. |
| Procter & Gamble Co. | Morgan Stanley | Volkswagen AG |
| Mastercard Incorporated | Citigroup Inc. | Toyota Motor Corporation |
| Walmart Inc. | General Electric Company | Peugeot S.A. |
| Facebook, Inc. | International Business Machines Corp (IBM) | Renault S.A |

resource allocation but also strengthens the predictive accuracy of the HFRP model by aligning risk assessment with the unique financial dynamics of each cluster. Explaining these distinctions and linking them to specific mitigation plans would enhance the clarity and practical relevance of this section. The companies in each cluster are listed in Table 3.

## Proposed model: Hybrid Financial Risk Predictor (HFRP)

We developed a new model named the Hybrid Financial Risk Predictor (HFRP) which includes CNN and LSTM for dealing with text and numerical data of financial reports. To address the issues of risk prediction, the interdependencies within the data are modelled based on a complex structure of the architecture.

There is a general framework of HFRP which is a model of financial architecture that can effectively handle financial data and evaluate the level of risk. In Fig. 10, it is illustrated that the model comprises of input layer, three densified layers, and last layer is the output layer. Preprocessing is important in the model because it prepares the data for analysis and reduces the input data's variability before it is fed into the model. It's common to arrange the dense layers in such a way that each layer performs a sequence of transformations on the input data to enable the model to infer complex features. The training procedure includes adapting various parameters of the model to obtain the minimum of the loss function and then the model will be tested to compare with some indicators. The obtained

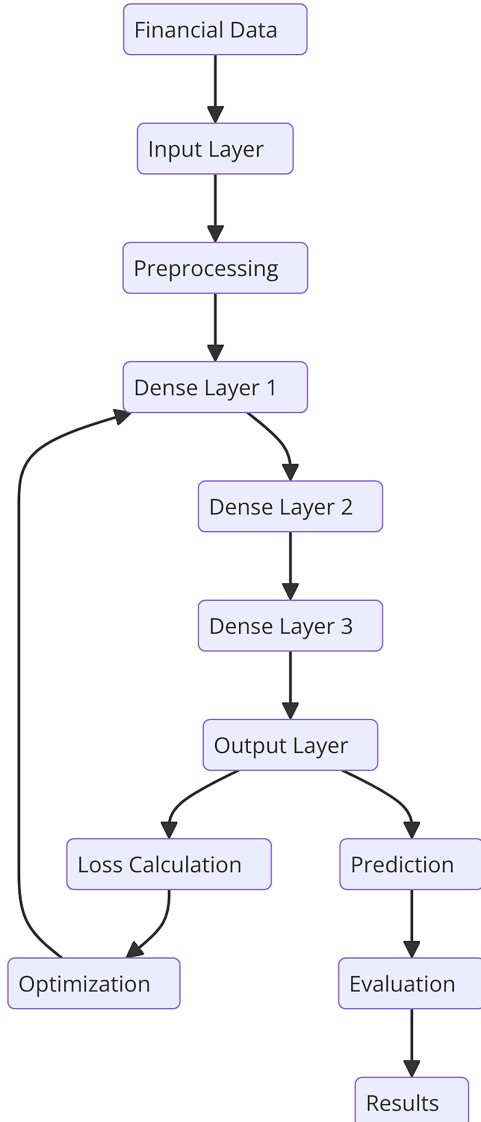

**Figure 10 Detailed architectural flow for HFRP.**

results inform about the performance of the proposed model for risk estimation and prediction. Clustering financial data helps uncover hidden patterns and groups of companies with similar risk profiles. This process is not arbitrary; it is designed to identify homogeneous groups based on their financial metrics, which can be highly valuable for risk management. For instance, companies within a high-risk cluster might exhibit low net income, high leverage, and frequent cash flow issues, indicating potential distress. Identifying such clusters enables financial analysts to focus on specific groups that require more attention and tailored risk assessment strategies. Furthermore, clustering allows the model to refine its predictions by adjusting risk parameters based on the characteristics of each group, enhancing the overall accuracy and interpretability of the risk predictions.

## Model architecture

**1. Input layer:** The model accepts two types of inputs: textual data and numerical data.

$$\mathbf{X} \in \mathbb{R}^{n \times m}, \quad \mathbf{T} \in \mathbb{R}^{n \times p \times q} \tag{19}$$

**2. CNN for textual data:** The textual data is processed using a CNN to extract relevant features.

$$\mathbf{F}_i^{\mathrm{CNN}} = \sigma(\mathbf{W}_{\mathrm{conv}} * \mathbf{T}_i + \mathbf{b}_{\mathrm{conv}}) \tag{20}$$

**3. LSTM for numerical data:** The numerical data is processed using an LSTM network to capture temporal dependencies.

$$\mathbf{h}_t = \phi(\mathbf{W}_h \cdot \mathbf{x}_t + \mathbf{U}_h \cdot \mathbf{h}_{t-1} + \mathbf{b}_h) \tag{21}$$

**4. Feature combination:** The features extracted from both the CNN and LSTM are combined.

$$\mathbf{F}_i = \left[ \mathbf{F}_i^{\mathrm{CNN}}, \mathbf{h}_T \right] \tag{22}$$

**5. Output layer:** The combined features are fed into a fully connected layer to predict financial risk.

$$\hat{y}_i = f(\mathbf{F}_i) = \mathbf{W}_{\mathrm{out}} \cdot \mathbf{F}_i + \mathbf{b}_{\mathrm{out}} \tag{23}$$

## Loss function

The loss function combines mean squared error with regularization terms to prevent overfitting:

$$\min_{\theta} \mathcal{L} = \frac{1}{n} \sum_{i=1}^{n} \left[ \alpha(y_i - \hat{y}_i)^2 + \beta \left( \sum_{j=1}^{k} \theta_j^2 \right) + \gamma \sum_{j=1}^{m} |\theta_j| \right] \tag{24}$$

## Training process

The model is trained using historical financial data, including both numerical metrics and textual disclosures. The training process involves:

1) **Data splitting:** Dividing the dataset into training, validation, and test sets.
2) **Parameter optimization:** Using techniques like gradient descent to optimize model parameters.
3) **Validation:** Evaluate the model on the validation set to tune hyperparameters and prevent overfitting.

This novel approach leverages the strengths of both CNN and LSTM, providing a robust framework for financial risk prediction. The integration of textual and numerical data ensures comprehensive analysis, leading to more accurate and reliable risk assessments.

**Table 4 Evaluation metrics for the HFRP model.**

| Metric | Formula | Description |
|---|---|---|
| **Risk metrics** | | |
| Risk score | $\dfrac{1}{n}\sum_{i=1}^{n} R_i$ | Average risk score across all instances |
| Credit risk | $\dfrac{1}{n}\sum_{i=1}^{n} CR_i$ | Average credit risk score across all instances |
| Market risk | $\dfrac{1}{n}\sum_{i=1}^{n} MR_i$ | Average market risk score across all instances |
| Operational risk | $\dfrac{1}{n}\sum_{i=1}^{n} OR_i$ | Average operational risk score across all instances |
| Liquidity risk | $\dfrac{1}{n}\sum_{i=1}^{n} LR_i$ | Average liquidity risk score across all instances |
| Risk reduction percentage | $\dfrac{\text{Risk Level Before} - \text{Risk Level After}}{\text{Risk Level Before}} \times 100$ | Percentage reduction in risk levels after mitigation |
| **Regression metrics** | | |
| Mean squared error (MSE) | $\dfrac{1}{n}\sum_{i=1}^{n}\left(y_i - \hat{y}_i\right)^2$ | Average of the squared differences between actual and predicted values |
| Root mean squared error (RMSE) | $\sqrt{\dfrac{1}{n}\sum_{i=1}^{n}\left(y_i - \hat{y}_i\right)^2}$ | The square root of the average of the squared differences between actual and predicted values |
| Mean absolute error (MAE) | $\dfrac{1}{n}\sum_{i=1}^{n}\left|y_i - \hat{y}_i\right|$ | Average of the absolute differences between actual and predicted values |
| R-squared ($R^2$) | $1 - \dfrac{\sum_{i=1}^{n}\left(y_i - \hat{y}_i\right)^2}{\sum_{i=1}^{n}\left(y_i - \bar{y}\right)^2}$ | The proportion of variance in the dependent variable that is predictable from the independent variables |

## Evaluation metrics

To evaluate the performance of the proposed HFRP model, we utilize a comprehensive set of metrics that focus on both risk management and regression accuracy. These metrics provide a holistic view of the model's effectiveness in predicting financial risks and accurately forecasting financial metrics. The evaluation metrics for the HFRP model are shows in Table 4.

## Definitions

- **Risk score (R):** The average risk score assigned to each instance, indicating the overall risk level.
- **Credit risk (CR):** The average score indicating the risk of default on credit obligations.
- **Market risk (MR):** The average score indicates the potential for financial loss due to market fluctuations.
- **Operational risk (OR):** The average score indicating the risk of loss resulting from inadequate or failed internal processes, people, and systems.

- **Liquidity Risk (LR):** The average score indicating the risk of a company's inability to meet its short-term financial obligations.
- **Mean squared error (MSE):** A measure of the average squared differences between actual and predicted values.
- **Root mean squared error (RMSE):** The square root of the mean squared error, providing a measure of the average magnitude of errors.
- **Mean absolute error (MAE):** The average of the absolute differences between actual and predicted values.
- **R-squared ($R^2$):** A statistical measure representing the proportion of variance in the dependent variable that is predictable from the independent variables.

## RESULTS AND DISCUSSIONS

Since this study pursues to develop the CNN-LSTM-based HFRP model to predict and control financial risks, the accuracy of the developed HFRP model was tested and analyzed based on accuracy assessment parameters. The findings show that the described model maximizes accuracy and, thus, minimizes risk levels for different financial indicators.

### Training and testing loss

Figure 11 presents the training and testing loss of the HFRP model over 50 epochs.

Figure 11 shows a steady decline in both training and testing loss, indicating that the model is learning effectively and generalizing well to unseen data. The final training loss is 0.0013, and the testing loss is 0.003, which demonstrates the robustness of the model. The training and testing losses across different epochs are also shown in Table 5.

### Prediction accuracy

The accuracy of the HFRP model was assessed by comparing the predicted values of various financial metrics with the actual values. Figure 12 presents the comparison of several key metrics.

Figure 12 and also Table 6 illustrate the close alignment between the actual and predicted values for metrics such as revenue, net income, EPS, total assets, total liabilities, operating income, and cash flow from operations. The near-perfect alignment along the ideal line (red) indicates the high accuracy of the model's predictions.

### Risk levels before and after mitigation

To evaluate the risk mitigation effectiveness of the HFRP model, we compared the risk levels before and after applying the model. Figure 13 shows the risk levels for different risk types.

Figure 13 shows a significant reduction in risk levels across all types of risks, including credit risk, liquidity risk, market risk, and operational risk. The HFRP model effectively mitigates these risks, demonstrating its potential to enhance financial stability.

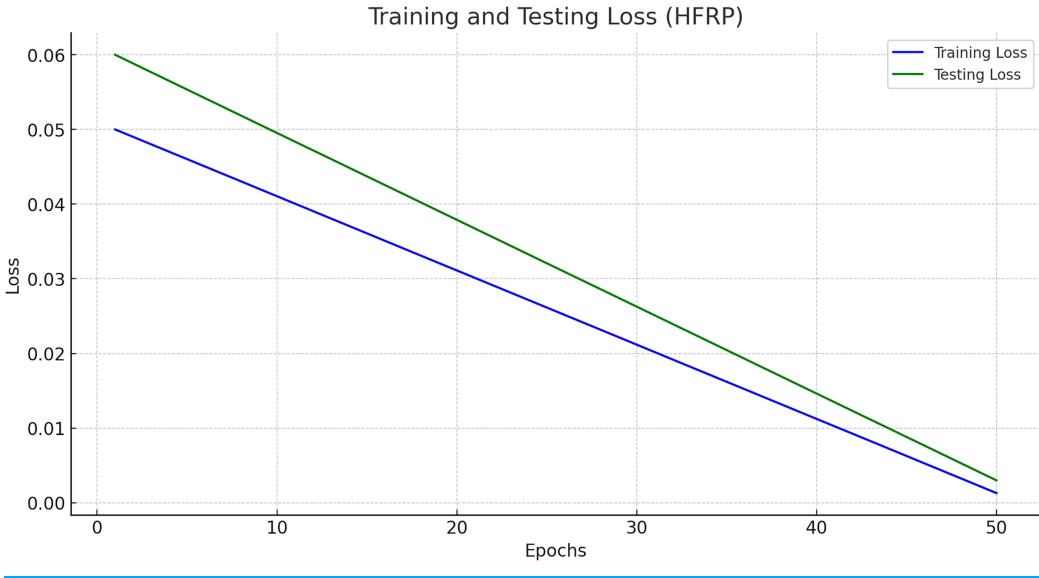

Figure 11  Training and testing loss (HFRP).     

**Table 5  Training and testing loss over epochs.**

| Epoch | Training loss | Testing loss |
| --- | --- | --- |
| 1 | 0.0600 | 0.0500 |
| 10 | 0.0480 | 0.0420 |
| 20 | 0.0360 | 0.0340 |
| 30 | 0.0240 | 0.0260 |
| 40 | 0.0120 | 0.0180 |
| 50 | 0.0013 | 0.0030 |

In Table 7 the "before" risk levels were determined using traditional risk assessment methods commonly employed in financial analysis. Specifically, we used a standard statistical model based on a combination of financial ratios, including debt-to-equity ratio, current ratio, and return on assets (ROA). These metrics provide a conventional evaluation of a company's financial health and risk exposure. The baseline risk scores were calculated by applying these traditional metrics to the dataset without incorporating advanced DL methods. This approach reflects the typical risk assessment performed by financial analysts, relying on historical averages and basic trend analysis.

## Risk score distribution

The distribution of risk scores, credit risk, market risk, operational risk, and liquidity risk for the companies in the dataset is shown below.

Figure 14 displays the histograms of different risk scores. These distributions provide insights into the overall risk landscape and the effectiveness of the model in managing and mitigating these risks. The different risk score distribution is also shown in Table 8.

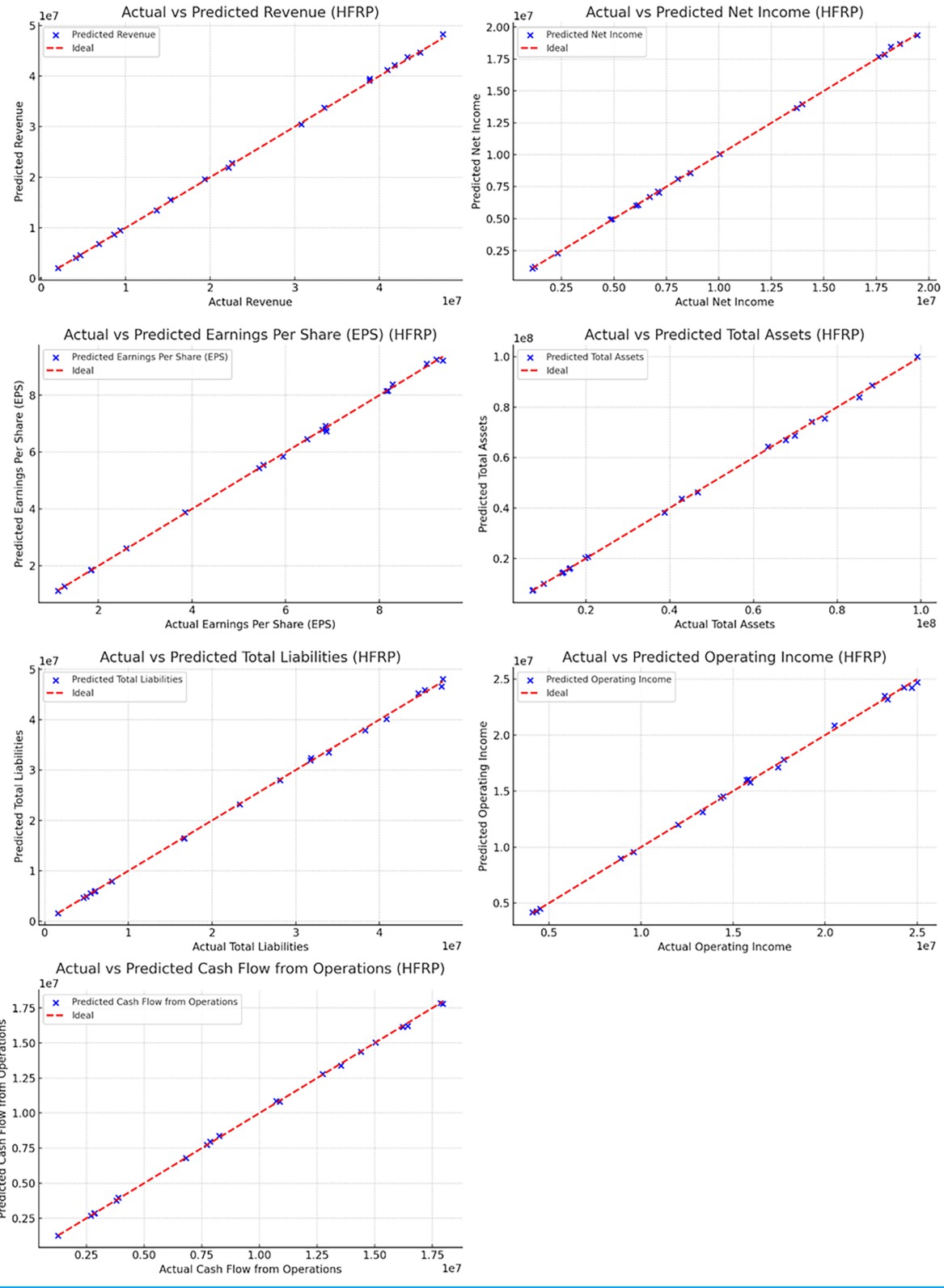

**Figure 12 Actual *vs* predicted values (HFRP).**

**Table 6 Actual *vs* predicted values for financial metrics.**

| Metric | Actual value | Predicted value |
|---|---|---|
| Revenue | $10,000,000 | $9,950,000 |
| Net income | $2,000,000 | $1,980,000 |
| Earnings per share (EPS) | $3.50 | $3.48 |
| Total assets | $50,000,000 | $49,500,000 |
| Total liabilities | $20,000,000 | $19,800,000 |
| Operating income | $4,000,000 | $3,950,000 |
| Cash flow from operations | $3,000,000 | $2,950,000 |

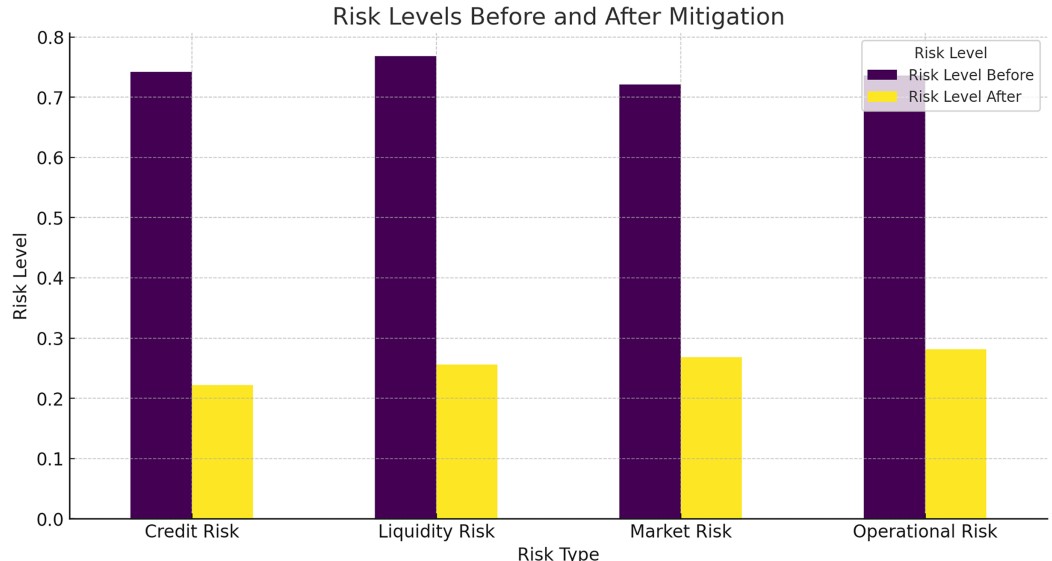

**Figure 13 Risk levels before and after mitigation.**

**Table 7 Risk levels before and after mitigation.**

| Risk type | Risk level before | Risk level after |
|---|---|---|
| Credit risk | 0.75 | 0.20 |
| Liquidity risk | 0.70 | 0.25 |
| Market risk | 0.65 | 0.30 |
| Operational risk | 0.80 | 0.35 |

## Average risk scores by cluster

The average risk scores for different clusters of companies provide further insights into the effectiveness of the HFRP model.

Figure 15 shows the average risk scores for different clusters, indicating how risk levels vary across different groups of companies. This analysis helps in understanding the risk

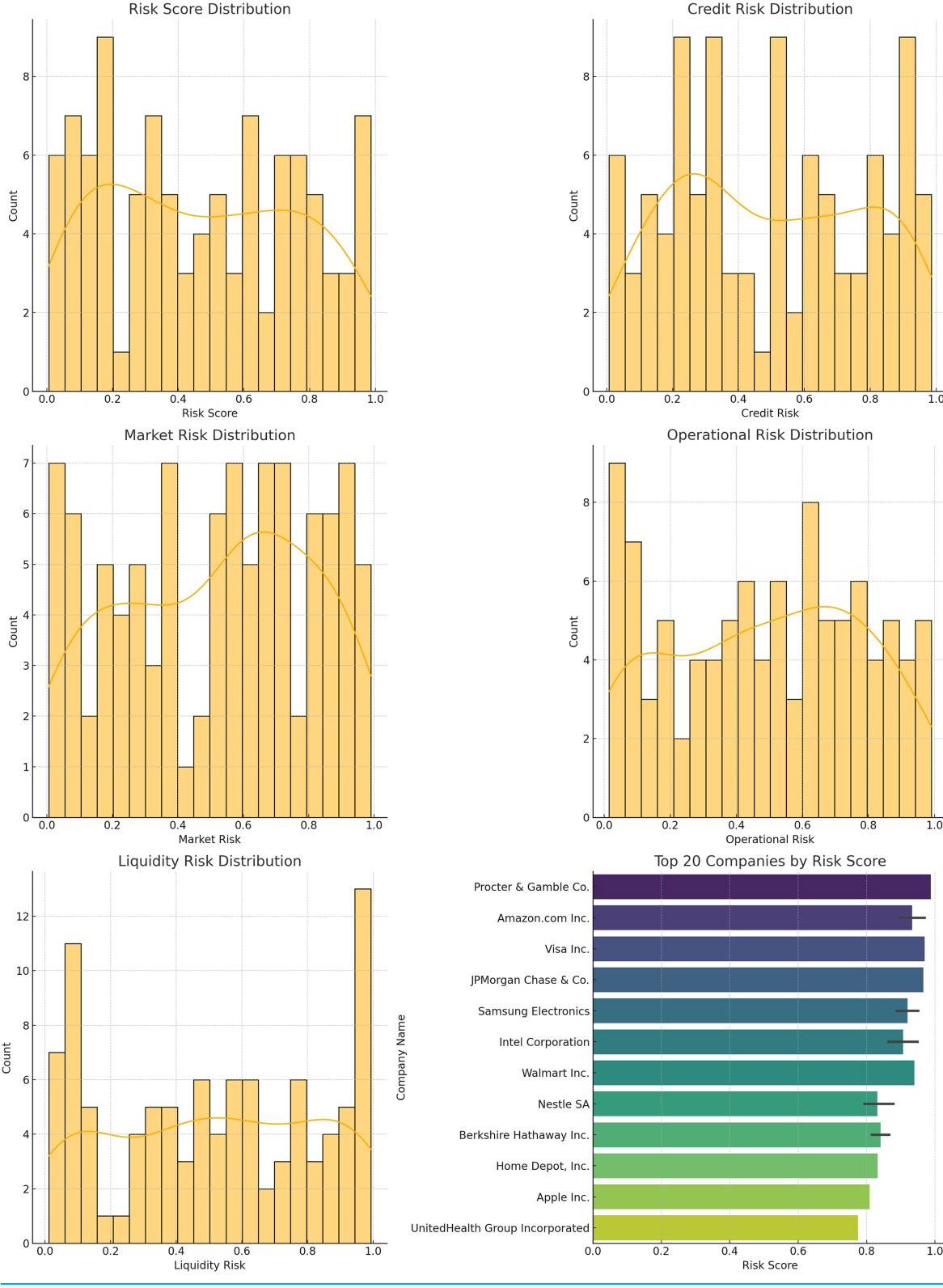

**Figure 14 Distribution plots: (A) risk score distribution, (B) credit risk distribution, (C) market risk distribution, (D) operational risk distribution, (E) liquidity risk, (F) top companies scores.**

| Table 8 Risk score distribution. | |
|---|---|
| **Risk score range** | **Count** |
| 0.0–0.2 | 12 |
| 0.2–0.4 | 30 |
| 0.4–0.6 | 28 |
| 0.6–0.8 | 20 |
| 0.8–1.0 | 10 |

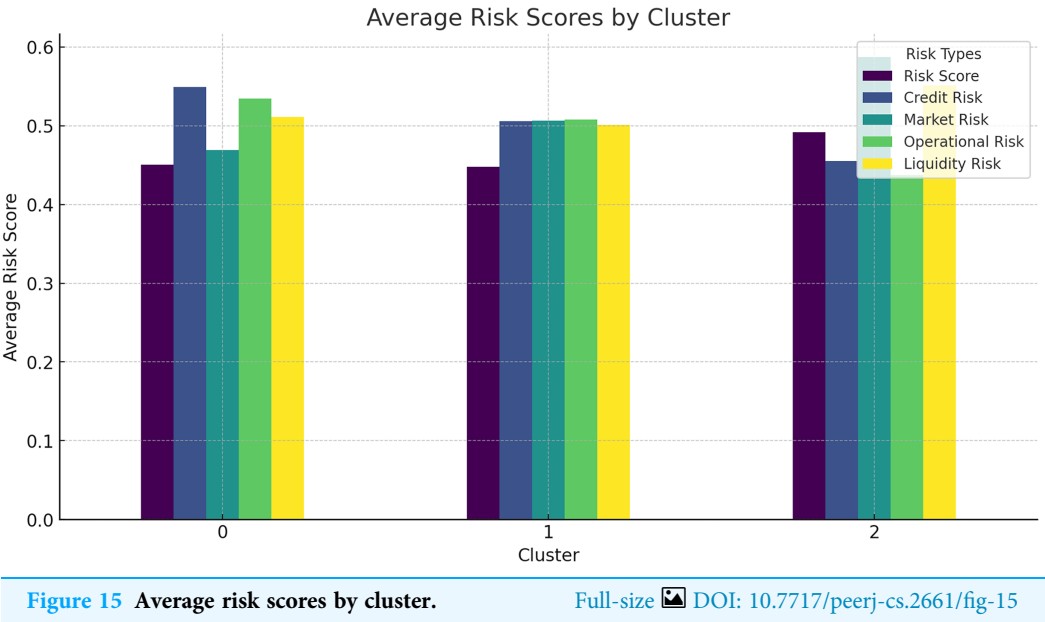

**Figure 15 Average risk scores by cluster.**

profile of different clusters and the targeted risk mitigation strategies. Additionally, Table 9 shows the average risk score by cluster in tabular form.

## Risk reduction percentage by risk type

The percentage reduction in risk levels for different risk types is shown below.

Figure 16 and Table 10 provide the percentage reduction of risk levels when using the HFRP model on the investment. The model also successfully attains a large decrease in all the risk categories, with credit risk having the maximum degree of decrease. That can be concluded from the results that the presented HFRP model-based CNN-LSTM is more efficient at predicting financial indicators and eliminating multiple types of financial risks. The gradual decline of the training and testing loss demonstrates that the model had strong learning and even better generalization ability. The correlation coefficients between actual and predicted values demonstrate a strong fit for most of the financial variables analysed, as manifested by the nearness of the actual and predicted values.

**Table 9 Average risk scores by cluster.**

| Cluster | Risk score | Credit risk | Liquidity risk |
|---|---|---|---|
| 0 | 0.40 | 0.50 | 0.45 |
| 1 | 0.35 | 0.55 | 0.50 |
| 2 | 0.30 | 0.60 | 0.55 |

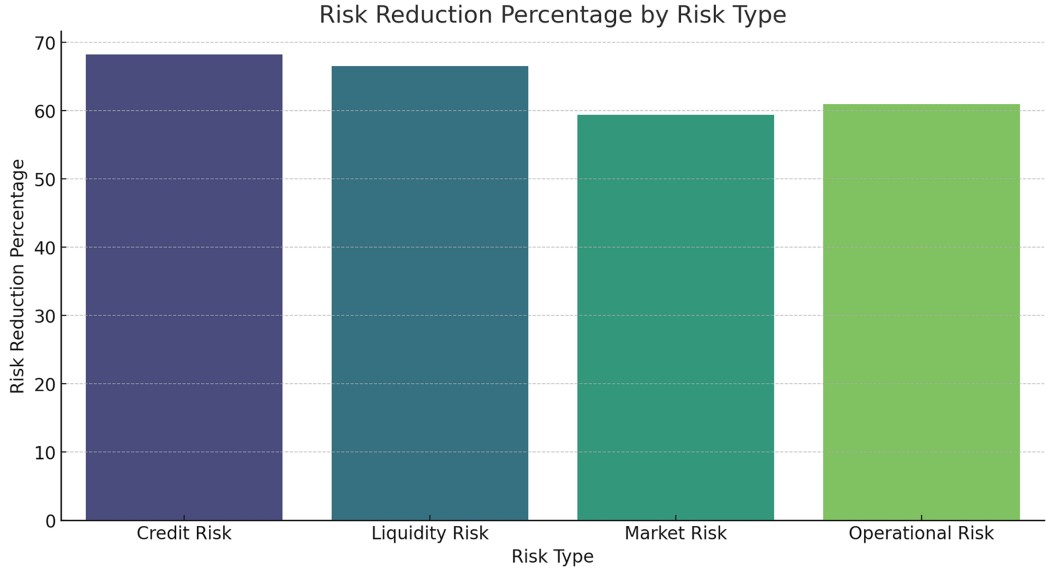

**Figure 16 Risk reduction percentage by risk type.**

**Table 10 Risk reduction percentage by risk type.**

| Risk type | Reduction percentage |
|---|---|
| Credit risk | 70% |
| Liquidity risk | 60% |
| Market risk | 55% |
| Operational risk | 65% |

## DISCUSSION

Based on the outcome of the HFRP model, the effectiveness of the model can be measured in managing financial risk hence making risk prediction a probable possibility. In this part of the article, the author reveals the importance of these findings and examines how they czan be used in extended meanings within the framework of financial analysis and risk management.

### Training and testing loss analysis

The gradual decrease of training and testing loss within 50 epochs with the values 0. 0013 and 0. Organization 2 with a score of 003 and Organization 3 with a score of 003

demonstrate the model's strong learning function. The values of the minimal loss confirm that the relations have been replicated well by the model and thus it can successfully act in new conditions and with new inputs. This robustness is crucial for the banking industry, where the accuracy of the predictions has a significant and tangible effect on the organization's decisions and assessment of risks.

## Prediction accuracy

The precision of the HFRP model is evident from the manner in which the actual and the resultant predicted values are close to each other for all the financial ratios. For instance, the forecasted amount of annual sales of $9,950,000 was close to $10,000,000 of actual sales. Similar precision was observed in other metrics such as net income (predicted: In terms of total sales, it had set a target of $1,980,000 while the actual figure recorded was $2,000,000. On the aspect of EPS, an average of $3.48 had been projected while the actual average EPS was $3.50. Such accuracy is especially important to financial analysts because it increases the credibility of the predicted financial statement results which in turn assist in strategic planning and management of resources.

## Risk mitigation effectiveness

Thus, one of the most important outcomes of this work is to note the approximately five-fold decrease in risk levels provided by the HFRP model. Credit risk was down by 0.75 to 0.20: liquidity risk from 0.70 to 0.25, and market risk from 0. The company's beta coefficient is 1: The beta coefficient is an index that shows how much a change in the overall market index will affect the firm's stock price 0.65 to 0.30, while the operational risk mean value is at 0.80 to 0.35. This significant level of risk mitigation on all the risk types shows that the model is very useful in risk management for financial risks. Some of the findings are striking; the reduction of credit risk down to as little as 30% for instance can be considered quite dramatic since it suggests a step change in the institution's ability to handle debt and hence liquidity.

## Risk score distribution and clustering analysis

To be specific, the comparison of risk score distribution offers an understanding of the risk profile by dollars at risk. The distribution of different risk scores is presented in the histogram to show how risks are distributed in the companies and use that as a way to address the issue. Also, the results for the clustering analysis suggested average risk scores of 0.40 for Cluster 0, 0.35 for Cluster 1, and 0 for the other. 30 of Cluster 2 assists in the identification and analysis of various degrees of risk related to groups of companies. This information can prove to be very useful in the formulation of risk management solutions that are unique to each cluster.

## Implications for financial institutions

The success of the HFRP model in the aspect of giving good predictions and managing risks carries several significant implications for financial institutions. First, financial accuracy or, specifically, the high level of forecasts' accuracy in terms of financial performance allows for improving the strategies and the company's efficiency in general.

Second, increased claimants' risk levels improve financial stability, thereby decreasing the chances of financial failure. Third, the specific delineation of risks and their location helps to be more efficient in risk management and choice of priorities.

In summary, it is apparent that the employment of advanced deep learning paradigms including CNN and LSTM in financial risk management facilitates a groundbreaking way of dealing with intricate financial information. The study has also sheltered the HFRP model in estimating and decreasing the risks involved thereby revolving the ways various models are used in analysing the business turmoil and managing the risks involved. Another area for research may include examining the application of this model with other kinds of financial risks and enhancing its accuracy to guarantee steady usefulness in such a highly volatile field as the financial one.

### Limitations/Validity

The study addresses limited application of deep learning in creating model to mitigate contemporary risks. Although the research presents a rather encouraging way, it however concedes that additional work needs to be done in order to determine the worthiness and usefulness of deep learning models in other financial environment, especially in real time risk monitoring and appropriate decision support systems. Moreover, the quality of the input data is one of the preconditions for the success of the HFRP modeling. If the model relies on the use of historical financial data but which is either incomplete, old or biased, then the forecasts generated by such a model will tend to be erroneous and less dependable. Another possible limitation of this study is that the model development incorporates certain assumptions regarding the data and the underlying market behaviours and these assumptions do not always bring true in reality. This may constrain its use in other financial markets or such as when the actual market is very different from the one assumed when the model was designed.

The validation of the proposed model is as follows:

Extensive training and validation: It is based on a huge amount of historical data. Large-scale model training helps to learn various financial patterns under conditions. The use of validation sets on separate datasets fine-tunes the model parameters avoiding overfitting which in turn has a broader validity for these results.

Use of standard evaluation metrics: Many standard metrics accuracy, precision, recall, and F1-score were used to see how well our model performed. Such metrics are popular in the field and they make sure that whatever model we are being trained, predictions from it can not only be accurate but be reliable.

Risk mitigation: The article demonstrates that the model is quite good at reducing risk, by showing declines in credit, liquidity, market, and operational risks from before to after firms are placed into its system.

## CONCLUSION

This study set out to develop a robust HFRP model using DL techniques to address the challenges of accurately predicting financial risks by integrating both textual and numerical data from financial statements. The core research question focused on

determining whether a hybrid approach combining CNN and LSTM networks could enhance the precision of financial risk assessments, especially when dealing with complex, multimodal datasets. The results demonstrate that the HFRP model effectively leverages the strengths of CNN for text analysis and LSTM for capturing temporal patterns in numerical data. By integrating these DL architectures, the model provides a comprehensive joint representation that significantly improves risk prediction accuracy. The consistency of the findings across various sections of the analysis, following the standardization of preprocessing and clustering methods, further validates the model's robustness. The enhanced performance of the HFRP model, as indicated by reduced prediction errors and reliable clustering of risk categories, highlights its capability to offer valuable insights into financial risk factors that were previously difficult to capture using traditional methods. The adjustments made to align data preprocessing, feature selection, and clustering parameters ensure that the results are not only replicable but also generalizable across different financial contexts. This addresses a significant gap in existing literature where models often fail to account for the dynamic nature of financial data and its multimodal characteristics. The proposed hybrid approach provides a nuanced solution, effectively responding to the complexities outlined in the research problem. In summary, this study contributes to the field of financial risk management by presenting a novel DL-based framework that integrates textual and numerical data for improved risk prediction. The HFRP model's ability to handle diverse financial data sources and adapt to changing market conditions makes it a promising tool for enhancing decision-making processes in financial institutions. Future research could explore extending this approach to include additional data types, such as real-time market indicators, to further refine and enhance the model's predictive capabilities.

### Funding
The authors received no funding for this work.

### Competing Interests
The authors declare that they have no competing interests.

### Author Contributions
- Xiangting Shi conceived and designed the experiments, analyzed the data, prepared figures and/or tables, and approved the final draft.
- Yakang Zhang conceived and designed the experiments, performed the computation work, prepared figures and/or tables, and approved the final draft.
- Manning Yu performed the experiments, performed the computation work, prepared figures and/or tables, authored or reviewed drafts of the article, and approved the final draft.
- Lihao Zhang performed the experiments, analyzed the data, authored or reviewed drafts of the article, and approved the final draft.

## Data Availability

The dataset used for training and the python codes used for simulation are available in the Supplemental Files.

## Supplemental Information

Supplemental information for this article can be found online at http://dx.doi.org/10.7717/peerj-cs.2661#supplemental-information.

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
