# Peer review of "Deep learning for enhanced risk management: a novel approach to analyzing financial reports"

_PeerJ Computer Science, doi:10.7717/peerj-cs.2661_

## Round 0.1 · original submission · Major Revisions

Major revisions are required as per the comments of the reviewers

Reviewer 1 ·

Basic reporting

The paper's language is clear but feels somewhat AI-generated. I'd recommend the author use generative AI more judiciously. The literature review is detailed, with each work introduced in separate paragraphs, which is commendable. However, it reads more like a list than a cohesive discussion, lacking organization and making it difficult for readers to see connections. The table comparing techniques is helpful. Consistent terminology, such as choosing either abbreviations or full names (e.g., DL vs. deep learning), is needed in table 1.

Experimental design

The research focuses on developing a Hybrid Financial Risk Predictor (HFRP) model using deep learning neural networks, aligning with the journal's scope. However, the research question isn't well-defined, and it's unclear how this work fills an existing knowledge gap. The paper includes numerous formulas and mathematical expressions in Section 3, but they lack explanations, making them difficult to understand. In Section 3, line 464, the phrase "combine DLR with traditional deep learning" is unclear because DLR is not explained (likely refers to deep reinforcement learning).

Validity of the findings

Sections like 3.6, where figures about data distribution are presented, only describe the axes without delving into the essential patterns and data dispersion. This lack of depth is also apparent in Sections 3.7, 3.8, and 3.9. In Section 3.10, the clustering process is described, but there’s no explanation of the purpose behind clustering, how the number of dimensions and clusters were chosen, or the financial characteristics identified. There are inconsistencies in the clustering results, particularly between the risk scores mentioned in sections 4.5 and 5.4 and those shown in Figure 12. These discrepancies need to be addressed to ensure the findings are reliable. Conclusions should be directly linked to the research question, providing clear insights rather than adding confusion.

Additional comments

The paper shows effort, but the presentation needs improvement. It feels like a collection of disconnected results rather than a coherent story. A strong research paper should answer key questions, not just list results without explanation. The author should focus on organizing their work around a clear narrative.

Reviewer 2 ·

Basic reporting

The paper presents a novel approach to financial risk management by introducing a HFRP model, which integrates CNN and LSTM networks to analyze financial reports. The model aims to improve accuracy in risk prediction and mitigation across various financial metrics, offering potential improvements over traditional methods.

1. Great literature search!
2. The introduction seems to be repetitive and could benefit from streamlining for clarity.
3. There is inconsistency in the terminology: some places refer to Long Short-Term Memory networks as "LST" and others as "LSTM." The authors should use the full name once and then consistently use one abbreviation.
4. The authors should consider moving some figures to supplementary materials to keep the main text concise, and emphasize the main story.
5. Each graph should be labeled with subtitles (e.g., Figure 2A, 2B, 2C, 2D).
6. The figure legends should be more descriptive to provide better context for the graphs.

Experimental design

1. It’s unclear how the risk levels were compared before and after applying the model in Figure 11. How was the “before” risk calculated? A detailed explanation is needed.
2. The clustering results show a mixture of industries within each cluster. The authors could explore which industries perform better with the model and provide potential explanations.
3. Data acquisition is not fully clear. The authors should cite the source of the data and include an example of a financial disclosure for clarity. The data sheet includes multiple entries for a single company, but lack of meaningful explanation on the differences.
4. It is unclear how the companies were selected for inclusion in the dataset.
5. It is recommended to deposit the code to a repository.
6. There is no mention of data normalization or how it was handled in the dataset.
7. More details are needed on how the authors conducted model tuning, including the choice of hyperparameters.

Validity of the findings

1. The authors claim a significant advantage of their model compared to others, but this could be strengthened by comparing the results to a baseline model.
2. In Figure 9, the training and testing loss do not seem to have plateaued, indicating that the model was still learning. The authors should explain the rationale for early termination of training.
3. The authors could enhance the validity by providing additional validation metrics such as accuracy and precision.

Additional comments

1. The bottom row of Figure 10 contains an empty plot.
2. In Figure 13, the last row plot shows data for only 12 companies, not 20, as indicated in the title.
3. In Figure 2 (bottom-right scatter plot), there appear to be more data points per company. The authors should explain this observation.
4. Overall, the manuscript is well-written. However, I spotted a few occurrences that there are grammar and punctuation errors. The authors should double-check the text.

---

## Round 0.2 · Minor Revisions

The paper need minor revisions!

Reviewer 1 ·

Basic reporting

The revised paper shows notable improvements, especially in the literature review, which now provides a clearer logical connection and context. The identification of the problem and knowledge gap is well-articulated. However, there are discrepancies in the model description that need addressing. The author claims that the HFRP model integrates CNN and LSTM to process both textual and numerical data, yet Figure 1 and other illustrations only depict a Multilayer Perceptron (MLP) with three dense layers. This inconsistency raises confusion about the model’s actual structure and capabilities. Additionally, the figure does not explain how textual and numerical data from financial reports are processed, as it only shows steps for handling textual data (e.g., tokenization, stop words removal, stemming). Reinforcement Learning and GANs, mentioned in Sections 3.5 and 3.6, are not reflected in Figure 1, leaving their integration into the model unclear. Including a table or paragraph explaining minor variables in the formulas, such as m, n, p, q, and k, would aid understanding, as their meanings are not immediately evident from context. Clarifying why Equation 5 includes penalty terms summing from 1 to k and m is also necessary for transparency.

Experimental design

The research question is now more clearly defined, addressing a specific knowledge gap within the scope of the journal. In Section 3.7, while the author describes figures showing dataset attributes, the analysis lacks depth; the paper should discuss the insights these figures provide, such as tendencies, skewness, outliers, and relationships, to strengthen the connection to the research goals. Similarly, Section 3.8 discusses frequent words in the textual data without explaining their potential benefits for analysis, which could be enhanced with specific examples. Section 3.9 should explore the insights gained from the correlation test and articulate how these insights contribute to achieving the research objectives. In Section 3.11, the discussion on PCA and clustering remains unclear; the paper should explain how the risk profiles and mitigation strategies differ among clusters, as the current figure (Figure 12) does not illustrate any differences.

Validity of the findings

While the model presents potential, the link between presented figures and the research goal remains weak. Sections 3.8 to 3.11 include figures without drawing substantive conclusions or connecting them to the research aim, which could leave readers questioning their purpose. Enhancing these sections with detailed interpretations and implications would strengthen the paper’s impact. The conclusions need to more explicitly tie back to the research question, delivering clear insights and resolving the initial inquiry.

Additional comments

The paper has made progress, but further refinements are necessary to enhance clarity and coherence. The author should focus on rectifying the discrepancies in the model's description, ensuring that figures substantively contribute to the research narrative, and providing detailed explanations for dataset handling. These improvements will significantly enhance the paper’s coherence, impact, and contribution to the field.

Reviewer 2 ·

Basic reporting

The revised version improved the overall quality of the manuscript. The authors addressed my comments very well. I recommend the manuscript for publication.

Experimental design

The authors provided additional explanations to clarify the methodology and analyses. They addressed my concerns thoroughly.

Validity of the findings

They addressed my comments thoroughly.

---

## Round 0.3 · accepted · Accept

The paper can be accepted. It was well improved!